# Cerebro-cerebellar networks facilitate learning through feedback decoupling

Ellen Boven[1,2,3], Joseph Pemberton[1,3], Paul Chadderton [2], Richard Apps [2] & Rui Ponte Costa [1] ✉

Behavioural feedback is critical for learning in the cerebral cortex. However, such feedback is often not readily available. How the cerebral cortex learns efficiently despite the sparse nature of feedback remains unclear. Inspired by recent deep learning algorithms, we introduce a systems-level computational model of cerebro-cerebellar interactions. In this model a cerebral recurrent network receives feedback predictions from a cerebellar network, thereby decoupling learning in cerebral networks from future feedback. When trained in a simple sensorimotor task the model shows faster learning and reduced dysmetria-like behaviours, in line with the widely observed functional impact of the cerebellum. Next, we demonstrate that these results generalise to more complex motor and cognitive tasks. Finally, the model makes several experimentally testable predictions regarding cerebro-cerebellar task-specific representations over learning, task-specific benefits of cerebellar predictions and the differential impact of cerebellar and inferior olive lesions. Overall, our work offers a theoretical framework of cerebro-cerebellar networks as feedback decoupling machines.

Learning ultimately depends on environmental feedback[1,2]. To learn efficiently animals and humans must make good use of this feedback to update their internal models of the world[3,4]. However, external sensory feedback is inherently delayed and incomplete, thereby reducing the rate and extent of learning in neuronal circuits[3]. These observations suggest that the brain may employ a general mechanism to facilitate learning when external feedback is not readily available.

The cerebellum is a region of the brain specialised in building predictive models[4,5]. In the classical view, the cerebellum learns predictive internal models on the motor domain[5–10]. Consistent with this view are a large body of experimental observations for which cerebellar dysfunction causes motor learning deficits. However, more recently, cerebellar dysfunction has also been associated with impaired language processing, cognitive associative learning and working memory[11–15]. An increasing body of behavioural[12,14,16–20], anatomical[21,22] and imaging[23] studies allude to a role of the cerebellum in cognition in animals and humans. Taken together, these studies suggest that the cerebellum learns internal models for both motor and non-motor functions in line with the proposed *universal functional* role of the cerebellum across the brain, including the cerebral cortex[9,24–26].

Despite growing experimental evidence there are no specific computational models aiming to capture the functional roles of cerebro-cerebellar interactions during learning of motor and non-motor tasks. Building on recent deep learning developments we theorise that the cerebellum predicts future cerebral feedback signals given current cerebral activity. This feedback predicted by the cerebellum is then sent back to the cerebral network to drive learning. Specifically, we model a given cerebral area as a recurrent neural network[27–30] which receives feedback predictions from a feedforward, cerebellar, network[6,7]. This view of cerebro-cerebellar interactions is in line with the classical forward models of cerebellar function[6,7], in that in our model the cerebellum makes forward predictions (i.e. generates cerebral feedback predictions) given current cerebral activity.

[1]Bristol Computational Neuroscience Unit, Intelligent Systems Labs, SCEEM, Faculty of Engineering, University of Bristol, Bristol BS8 1TH, UK. [2]School of Physiology, Pharmacology and Neuroscience, Faculty of Life Sciences, University of Bristol, Bristol BS8 1TH, UK. [3]These authors contributed equally: Ellen Boven, Joseph Pemberton. ✉e-mail: rui.costa@bristol.ac.uk

We test our model on a range of sensorimotor, pattern recognition and visual-language tasks. Using these tasks we demonstrate that cerebellar feedback predictions conveyed to the cerebral cortex facilitate learning. Moreover, models without a cerebellar component exhibit slower learning and dysmetria-like behaviours, consistent with a wide range of behavioural observations[11,14,31,32]. Our results indicate that the cerebellar-mediated facilitation of cerebral learning relies on the ability of the cerebellum to provide effective cerebral feedback predictions. Finally, we make several experimentally testable predictions regarding cerebro-cerebellar representations, task-specific temporal feedback, cerebro-cerebellar activity coupling and the different contributions of cerebellar output and inferior olive for task learning.

## Results

### A systems-level computational model of cerebro-cerebellar interactions

In order to understand how cerebellar computations may shape cerebral processing, we introduce a cerebro-cerebellar systems-level model based on a recent deep learning algorithm[33]. In line with previous work we model a given cerebral cortical area $A$ as a recurrent neural network (RNN)[27–30] which is coupled with a cerebellar module $C$ − cerebro-cerebellar RNN (ccRNN). We model the cerebellar module as a simple feedforward network $C$ (Fig. 1a) in line with the cerebellar architecture[6,7,9]. The input layer of the cerebellar network mirrors Granule cells (GC), and receives cortical activity $\mathbf{a}$. The output layer models Purkinje cells (PC) and provides cerebellar predictions back to the cerebral cortex (see the "Methods" section). To capture the dimensionality expansion observed between cerebral and cerebellar networks we constrain our model with $M \gg N$, where $M$ corresponds to the number of GCs, $N$ the number of cerebral neurons and use the same ratio found experimentally $\frac{M}{N} \sim 4$[34,35].

We study the behaviour of our model in a range of tasks. To train the model we use a prediction error function $E_{\text{task}}$ which compares the model output with task-specific external feedback. Using standard gradient descent methods we generate feedback signals of a specific temporal horizon (see example of a RNN unrolled in time in Fig. 1b), $\mathbf{fb}_t$, which is then used to update the RNN input and recurrent weights

(Fig. 1a; see the "Methods" section). For computational efficiency and in line with previous models we use a time-discrete approximation of time-continuous RNN models[28].

Following our theoretical proposal, the cerebellar module $C$ learns continuously to predict cerebral feedback $\mathbf{fb}_t$ given cerebral cortical activity $\mathbf{a}_t$. The cerebellar network is optimised through error signals computed by comparing the actual cerebral feedback $\mathbf{fb}_t$ at time $t$ with the cerebral feedback predicted by the cerebellum $\hat{\mathbf{fb}}_t$. We postulate that this comparison is done in an inferior olive-like structure, $E_t^C = ||\mathbf{fb}_t - \hat{\mathbf{fb}}_t||^2$, that generates error signals which are used to optimise the cerebellar network (see the "Methods" section). However, similar to external feedback, actual cerebral feedback is not always available, which would impact the ability of the cerebellar network to learn online to produce effective feedback signals. To circumvent this problem we propose that the cerebellum learns using its own feedback predictions when cerebral feedback is not available (Fig. 1b)[33]. This leads to the following target feedback $\overline{\mathbf{fb}}_t \sim \mathbf{fb}_t + C(\mathbf{a}_{t+1})$ where $\mathbf{fb}_t$ is the true cerebral feedback and $C(\mathbf{a}_{t+1}) = \hat{\mathbf{fb}}_{t+1}$ is a self-prediction term that enables the cerebellum to learn online (see full details in Methods). Learning by self-prediction (bootstrapping) is commonly used in reinforcement learning and is of key importance in our model for the cerebellum to learn to provide effective cerebral feedback predictions.

### Cerebro-cerebellar model facilitates learning in a simple sensorimotor task

Inspired by classical sensorimotor studies in the cerebellum, we first test a simple visuomotor task[11,31,32,36,37]. In this task, the model must draw a straight line in a two-dimensional space, expressed in $x$, $y$ coordinates, towards one of seven target locations given a target-specific cue at the start of the task (Fig. 2a, top). We train a cerebro-cerebellar RNN (ccRNN) and a cerebral-only RNN (cRNN) to perform this task (see full details in "Methods" section). To train the models we provide teaching feedback, by comparing the cerebral network output with the optimal trajectory (i.e. a straight line between starting and end points; Fig. 2a), where both feedback and model output are expressed as $x$, $y$ coordinates (but we observe similar outcomes using a more realistic point-mass model for the output, Fig. S2). In addition, this

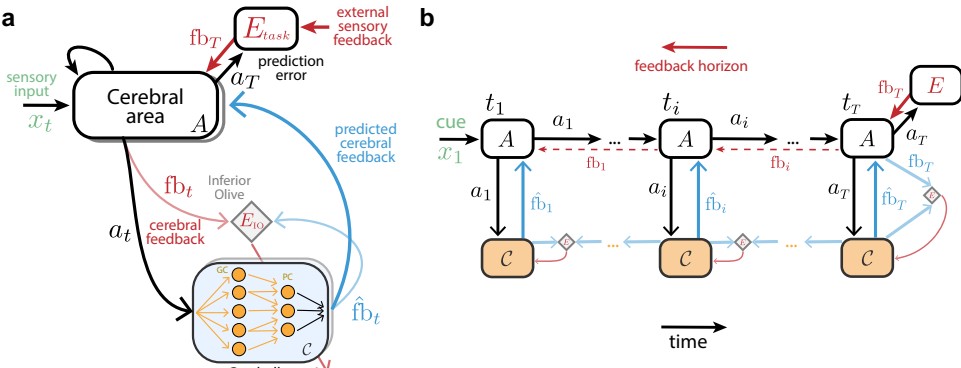

**Fig. 1 | Cerebro-cerebellar networks as feedback prediction machines.**
**a** A recurrent cerebral cortical network $A$ learns through external sensory feedback given by a task-specific prediction error module $E_{\text{Task}}$ computed at the end of a task $\mathbf{fb}_T$ (top red arrow). The cerebellum aims to continuously predict the feedback expected by the cerebral network $\hat{\mathbf{fb}}_t$ (blue) given current cerebral activity $a_t$ (black). The cerebellar network (i.e. granule cells; GC and Purkinje cells; PC) learns through prediction errors (bottom red arrow) computed at the inferior olive (diamond) by comparing predicted cerebral feedback $\hat{\mathbf{fb}}_t$ with actual cerebral feedback $\mathbf{fb}_t$ (light blue). Shaded boxes represent multiple cerebral areas and cerebellar modules that may be interacting in parallel (see Fig. S1 for the same framework applied to decoupling across multiple brain areas). **b** Example of cerebro-cerebellar model unfolded in time in which the cerebral network learns to associate

a cue given at $t_1$ ($x_1$, green) with feedback received at the end of the task, $t_T$ (cf. Fig. 2). At the end of the task the cerebral network $A$ receives external sensory feedback $\mathbf{fb}_T$ (red), which is transmitted to the cerebellar network as cerebral feedback $\mathbf{fb}_T$ (light blue). Here we highlight a case of cerebral feedback horizon stopping at the end of the task $T$, but feedback may also be available earlier in the task (dashed red arrows). The cerebellum generates cerebral feedback predictions $\hat{\mathbf{fb}}_T$ (blue) given cerebral activity $a_T$ (black) and learns using inferior olive (diamond) error signals (red arrow). Before $t_T$ cerebral feedback may not be readily available, thus the cerebellum learns through self-predictions. In this case the inferior olive (diamond) compares old cerebellar predictions (e.g. $\hat{\mathbf{fb}}_i$) with the new one (e.g. $\hat{\mathbf{fb}}_T$) to generate cerebellar learning signals (red arrow; see main text and "Methods" section for details).

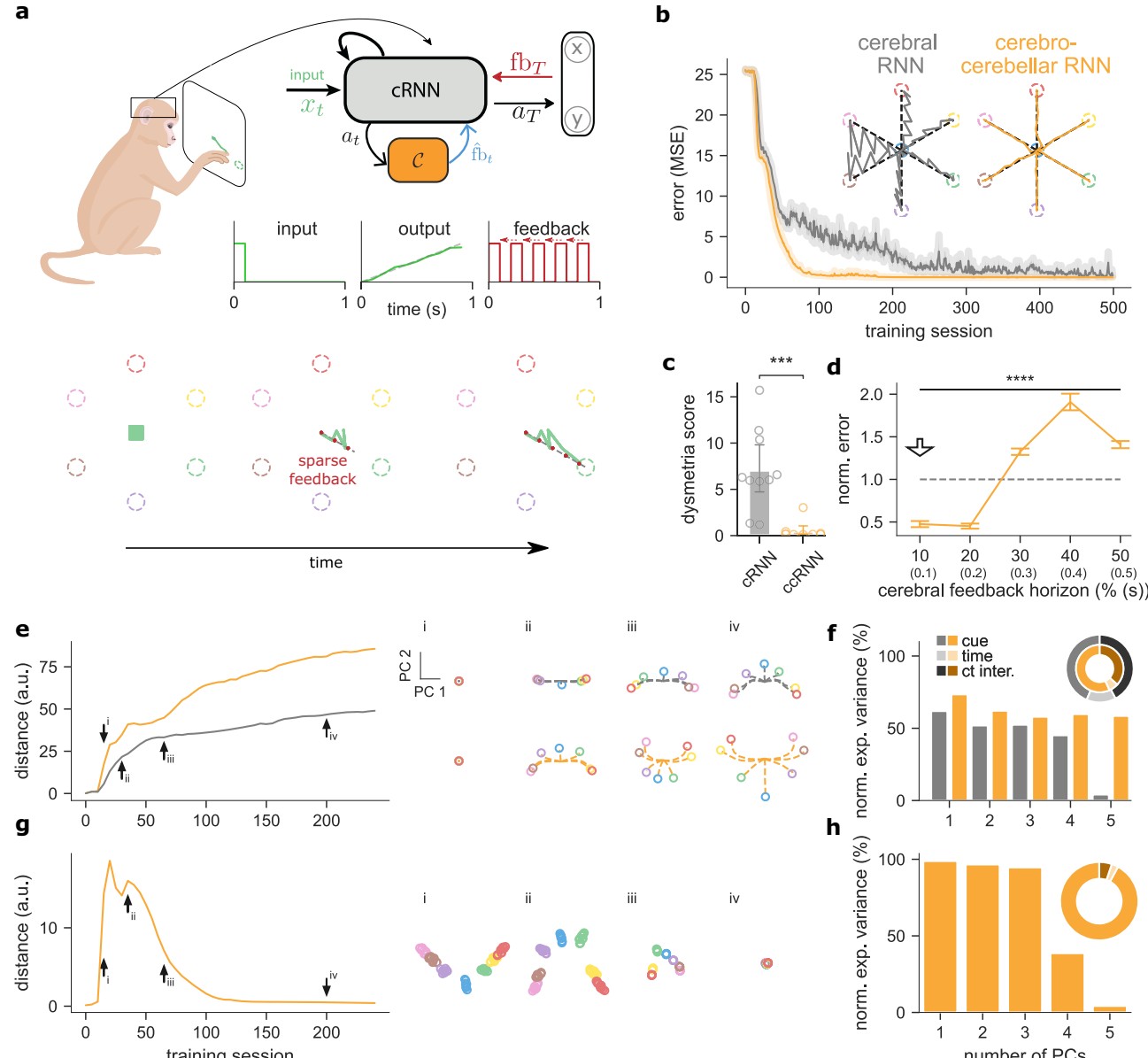

**Fig. 2 | Cerebro-cerebellar model improves learning in a simple line drawing sensorimotor task. a** Schematic of a macaque monkey performing a simple line drawing task (top left). A cerebro-cerebellar RNN (ccRNN) in the macaque's brain receives cue-specific input and learns to produce the desired trajectory (top right). The temporal profile of input, output (dashed gray line represents the target trajectory) and feedback are also shown (bottom right). There are six possible target lines (coloured dashed circles; plus a 7th target for which the model must remain still) and feedback (dashed gray line) is provided at a regular interval (bottom; see the "Methods" section). In the example shown the model must draw a straight line toward the green target. **b** Error between model output and desired target trajectories for cerebellar RNN (gray, cRNN) and cerebro-cerebellar RNN (orange, ccRNN). Insets: Model trajectory produced for all cues after learning. **c** Dysmetria score for cRNN and ccRNN. The dysmetria score quantifies how smooth the movement is after learning (see the "Methods" section). **d** Normalized model mean squared error after learning for different cerebral feedback horizons. Feedback horizon is denoted as a percentage of the total task sequence. Arrow indicates the feedback horizon used by the cerebral network in the other panels. **e** Euclidean distance between the cerebral RNN dynamics corresponding to the two leading cue-specific principal components. Results are given for both the cRNN

(grey) and ccRNN (orange) models. Arrows highlight training sessions of cue-specific demixed principal components (dPCs) plotted on the right for early (i), early-mid (ii), mid (iii) and late (iv) learning, for both cRNN (top) and ccRNN (bottom). Dashed lines represent the trajectory of the 2D neural dynamics throughout the task (circle represents last timestep). **f** Normalised cue-specific explained variance of the RNN for both cRNN (gray) and ccRNN (orange). Circular plot shows the total explained variance for cue (medium-dark colours), time (light colours) and cue-time interaction (dark colours) task variables. **g** Euclidean distance of the cue-specific two-dimensional neural activity for the cerebellar network (orange, ccRNN model). Arrows indicate training sessions highlighted on the right (i–iv) as in (**e**). In contrast to the cerebral network (**g**) here there is no task trajectory encoded − multiple circles represent the temporal points during the task. **h** Normalised explained variance for cue-specific dPCs of the cerebellar network. Circular plot colour-coding same as (**f**). ***$p < 0.001$, ****$p < 0.0001$ (two-sided paired $t$-test between cRNN and ccRNN). Error bars represent mean ± SEM across 10 different initial conditions. The animal drawing used in (**a**) is available at https://www.scidraw.io/drawing/445 with a Creative Commons license (https://creativecommons.org/licenses/by/4.0/). Error bars represent mean ± SEM across 10 different initial conditions. Source data are provided as a Source Data file.

feedback is delayed with respect to the initial cue and incomplete (i.e. only available every few time steps). This setup models a more realistic setting in which task feedback is not always readily available. When this feedback is available at time $t$ we calculate the prediction error as $E_{task} = ||\mathbf{l}_t - \hat{\mathbf{l}}_t||^2$, where $\mathbf{l}_t$ is the desired two-dimensional trajectory (i.e. set of feedback points; cf. Fig. 2 schematic) and $\hat{\mathbf{l}}_t$ is the current model output given by a linear readout on the network activity $\mathbf{a}_t$ (see the "Methods" section). Here we consider a feedback interval at every other time step for both cRNN and ccRNN (but see below for more general cases).

During learning the ccRNN model achieves near-zero error after a relatively small number of training sessions, while the cRNN, which lacks the cerebellar component, also learns but more slowly and with higher variability (Fig. 2b). These observations are in line with a large body of cerebellar experiments[11,31,32]. In addition, we observe differences at the level of model output trajectories. While the ccRNN produces smooth and straight trajectories, the cRNN displays a much more variable trajectory towards all targets (Fig. 2b). Due to the sparse task feedback in the absence of a cerebellar network, the cRNN is not able to learn a correct trajectory in points for which there is no direct feedback thus overshooting the target trajectory. In cerebellar patients, this effect is referred to as dysmetria[38] which in the motor domain results in ataxia. Ataxia is the lack of coordination and fine control during voluntary movements, a defining symptom resulting from cerebellar malfunction[11,38]. To evaluate the degree of dysmetria-like output in our models we measure the error between the model output and the optimal trajectory (i.e. a straight line in this case; see the "Methods" section). When applying this measure, the ccRNN shows a clear reduction in ataxia-like behaviour compared to cRNN (Fig. 2c). Finally, we demonstrate the benefits of our model compared to classical models in solving tasks of a temporal nature. We trained both an Albus–Marr feedforward model and a model with a fixed RNN on drawing tasks. Due to their inability to perform temporal credit assignment, both models fail to learn, and thus do not exhibit the same properties as the ccRNN (Fig. S3).

To highlight the conditions for which the cerebellum may facilitate learning in cerebral networks we test different lengths of cerebral feedback horizon (see the "Methods" section). Our results show that the ccRNN only facilitates learning for short to medium feedback horizons (<50%, Figs. 2d, S4). These results suggest that the cerebellum is particularly important for cerebral learning in conditions in which cortical networks do not have internal effective feedback available for learning. This is consistent with experimental observations showing that the cerebellum becomes more important in the presence of challenging task conditions for which cerebral feedback might be short[39]. In contrast, for long cerebral feedback, having a cerebellar module harms learning. In this case, the cerebral network has the level of feedback required to learn effectively, thus the noise inherent in the cerebellar feedback can impair learning. This observation suggests that the brain may use intermediate brain structures, such as the thalamus and the pons to gate cerebro-cerebellar interactions depending on task properties (see Discussion).

Next, to gain insight into how cerebral and cerebellar neuronal representations evolve jointly during learning, we use a dimensionality reduction method (demixed principal component analysis (PCA); see the "Methods" section). Demixed PCA (dPCA) enables us to extract low-dimensional neuronal representations that capture maximum variance across task variables. First, we focus on the two most informative cue-specific principal components using the neural activities of the recurrent neural network for both cRNN and ccRNN (see all components in Figs. S5–7). Next, we calculated the two-dimensional Euclidean distance across the seven different possible cues at each timestep (see the "Methods" section). Our results show that the ccRNN

cerebral network is characterised by a stronger increase in the separation of stimulus components over learning when compared to the cRNN cerebral network (Fig. 2e). To contrast task-specific components with general temporal information, we compare the level of cue-specific and time-specific explained variance in both models. In order to directly compare the cue-specific explained variance of each component we normalise by the variance of each component for the respective model. Overall, ccRNN captures more cue-specific explained variance when compared with cRNN (Fig. 2f), which demonstrates that ccRNN encodes more task-relevant information in a task that requires the model to associate the cue information with specific output trajectories. Next, we applied dPCA to the activity of cerebellar neurons. Since the cerebellar module facilitates cue-to-target learning we expected cerebellar representations to be mostly dominated by task-specific information. This is indeed what we find, our results show that the distance between cue-related components is stronger during periods of high learning (Fig. 2g; compare with Fig. 2b; similar to a linear regression analysis, Fig. S8) and that most of the variance is explained by cue-specific dPCs (95.4%; Fig. 2h).

Overall, our results suggest that in the context of a simple sensorimotor task, cerebellar-mediated decoupling of cerebral feedback enables faster learning and smoother motor trajectories. In addition, it makes a number of experimentally testable predictions about the evolution of task-specific cerebro-cerebellar representations throughout learning.

## Cerebro-cerebellar model improves learning in complex sensorimotor and discrimination tasks

Under naturalistic conditions, both animals and humans have to learn complex sensorimotor transformations[40,41]. To test whether the results from the simple visuomotor task generalise to more realistic settings we explore a range of more advanced sensorimotor tasks. In contrast to the previous task in which sensory input (i.e. the stimulus) was only provided at the start of the task, in these tasks the model receives a constant stream of external input. In particular, ordered segments (i.e. a row of 28 pixels; see the "Methods" section) of a handwritten digit from the MNIST dataset (see the "Methods" section) are provided as input and the model has to simultaneously draw a shape associated with the digit (see a few MNIST samples in Fig. S9). We refer to this task setting in which input is provided over time as *online*. Given this input, we consider two task variants (Fig. 3a) in which the model has to either draw a corresponding (i) straight line (online line drawing (LD) visuomotor task) or (ii) non-linear trajectory (online digit drawing (DD) visuomotor task). Both tasks provide a more realistic model of drawing tasks (cf. Fig. 2) in which lines must be drawn given complex continuous sensory input. As in the previous task, we consider cases of sparse task feedback.

As in the simple visuomotor task, here the ccRNN learns faster (Fig. 3b; across different stimulus noise levels, Fig. S10) than cRNN while showing a strong reduction in dysmetria-like trajectories (Fig. 3c). The ccRNN also facilitates learning when in the presence of short to medium feedback horizon in the cerebral network (Fig. 3d). Moreover, our model predicts that if the sensory input is compressed in time, then the need for temporal credit assignment and therefore a cerebellar module is reduced (Fig. S11).

There is growing evidence suggesting that the cerebellum is also involved in non-motor tasks[12,14,16–20]. To test whether our observations in the sensorimotor tasks generalise to non-motor domains while using similar input statistics as the previous tasks we trained the model in a visual discrimination task. In this task, the model receives the same handwritten digits presented sequentially over time but now must discriminate between the 10 classes of digits (online visual discrimination task, Fig. 3a) and only receives external feedback at the

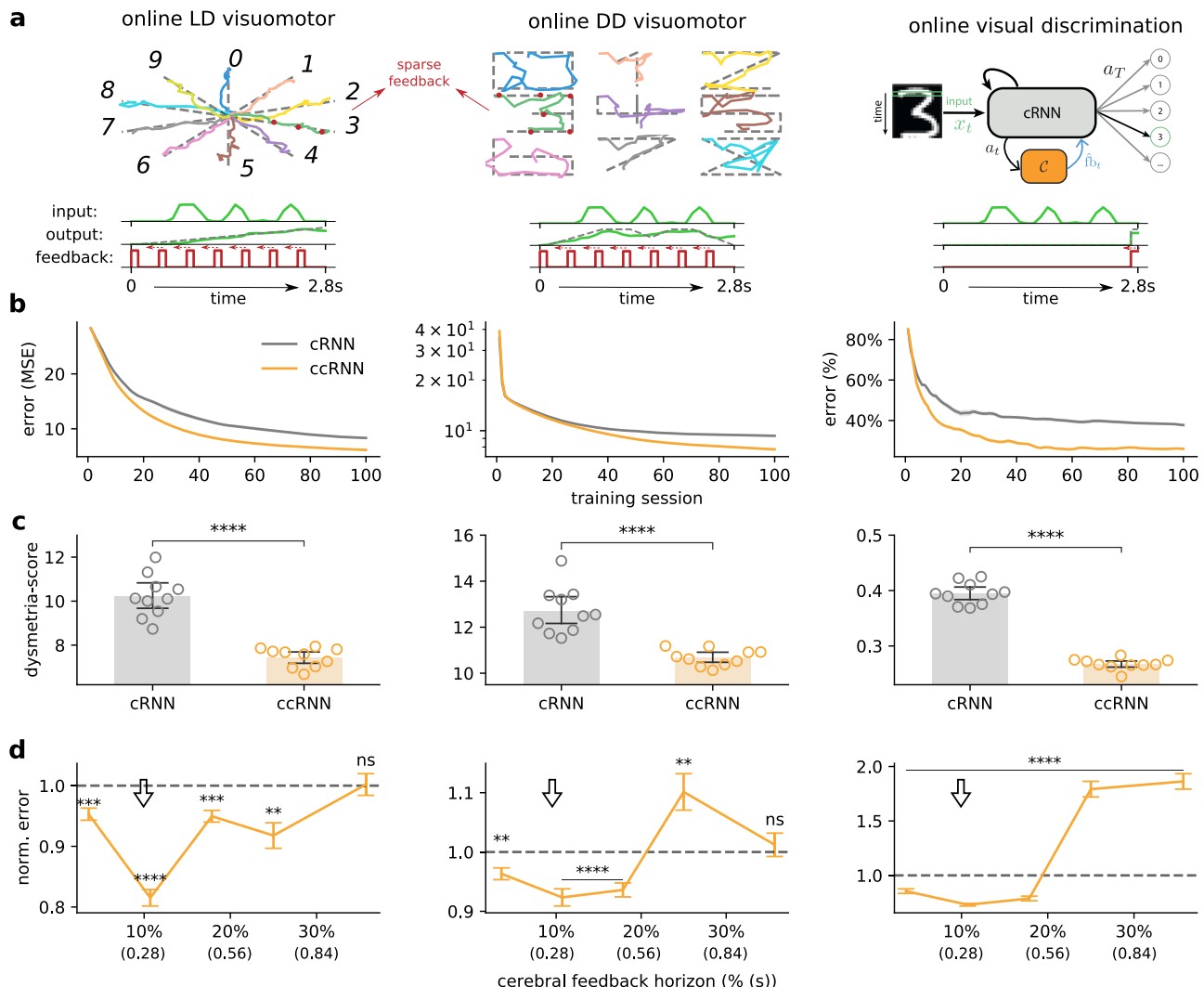

**Fig. 3 | Cerebro-cerebellar model improves learning in online complex sensorimotor and sensory discrimination tasks. a** Model behaviour across three tasks using a dataset of handwritten digits, each presented sequentially to the network (Methods and main text). Online line drawing (LD) visuomotor task: given temporally varying visual input the model is trained with sparse feedback (red dots) to draw a straight line (top left). Online digit drawing (DD) visuomotor task: given temporally varying visual input the model is trained to draw a digit following a template (top middle); target trajectories are in dotted grey and model input/output is coloured by digit. Online visual discrimination task: pattern recognition variant in which the model is trained to discriminate between 10 different digits given as sequential input. A representation of the structure of the input (green), output (green; target in grey) and feedback (red) for each task is also given (bottom of each task). **b** Learning curves for the three tasks for both cerebral RNN (gray, cRNN), cerebro-cerebellar RNN (orange, ccRNN). The cerebral network in all tasks uses approximately cerebral feedback horizon of 10% (cf. **d**). **c** The dysmetria score quantifies the irregularity in movement during the testing phase of the model (online LD and DD visuomotor tasks) or the uncertainty in the sensory discrimination (online visual discrimination task). **d** ccRNN model performance relative to cRNN across different degrees of cerebral feedback horizon (ns denotes not significant: $p = 0.921$ in the online LD visuomotor and $p = 0.567$ in the online DD visuomotor). Arrow indicates the feedback horizon used in (**b**, **c**). **$p < 0.001$ ***$p < 0.0001$, ****$p < 0.0001$ (two-sided paired t-test between cRNN and ccRNN). Error bars represent mean ± SEM across 10 different initial conditions. Source data are provided as a Source Data file.

end of the input presentation. In line with the results in the visuomotor tasks, we find that ccRNN also facilitates learning in this task, achieving higher accuracy after only 10 training sessions (Fig. 3b). Here we use the certainty the model has about the current class as a measure of dysmetria of thought[42] (see the "Methods" section). Similarly to the tasks above, we find that dysmetria-like behaviours are reduced in the ccRNN model, which in this case shows that the model produces more accurate decisions (Fig. 3c). In line with previous tasks, a cerebellar module facilitates learning in the presence of weak cerebral feedback (Figs. 3d, S12). Finally, we have also used this task to highlight the importance of cerebellar learning by self-prediction (i.e. bootstrapping; Fig. S13). These results are in line with the growing number of studies implicating the cerebellum in sensory discrimination and decision-making tasks[19,43,44].

## Cerebellar-mediated learning facilitation depends on task feedback interval

In sensorimotor tasks, there are physiological constraints inherent to animals and humans which impose limits on the rate at which external sensory feedback is available[45–47]. To determine the rate of external feedback for which cerebellar predictions are most valuable we trained the model in two tasks (simple LD and LD visuomotor tasks) with a range of external feedback intervals. This feedback interval defines the rate at which external feedback is available for learning, resembling sensorimotor feedback which is typically sporadic rather than continuous[11,48,49]. We find that when external feedback is given at short intervals there is little advantage of the feedback predictions from the cerebellar component for both the simple LD and online LD visuomotor tasks, which is evident in dysmetria score (Fig. 4a, b) and

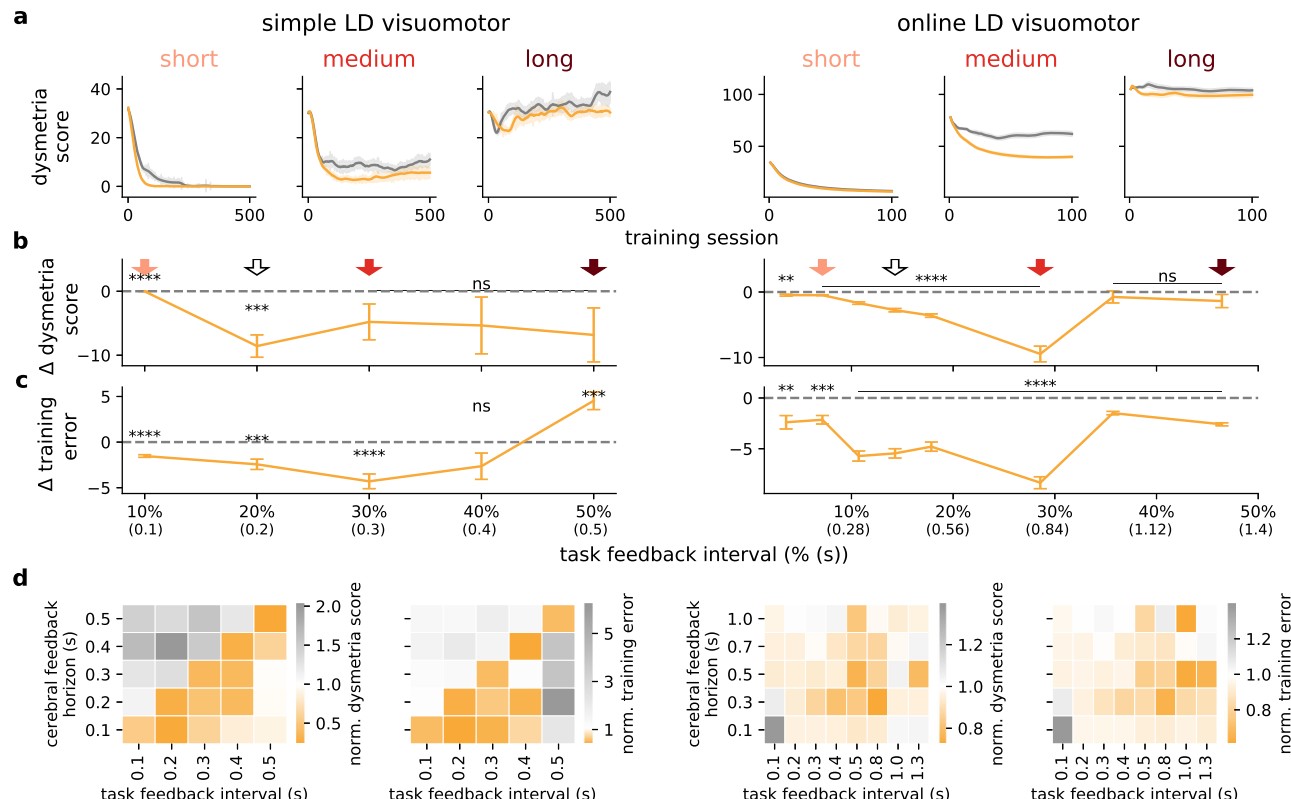

**Fig. 4 | Cerebellar-mediated facilitation of learning depends on task feedback interval. a** Dysmetria score during learning for short (light red), medium (red) and long (dark red) levels of feedback interval for the simple and online LD visuomotor tasks and both models cRNN (gray) and ccRNN (orange). Degrees of redness **b** Difference in dysmetria score between ccRNN and cRNN for varying degrees of task feedback intervals (ns denotes not significant: $p = 0.122$ (30%), $p = 0.268$ (40%), $p = 0.142$ (50%) for simple LD and $p = 0.444$ (36%), $p = 0.209$ (46%) for online LD). Degrees of red in arrows indicate the respective interval as in (**a**) while the white arrow indicates the feedback interval used in Figs. 2 and 3. Task feedback interval is given as a percentage of the total task time. **c** Difference in training error between cRNN and ccRNN for varying degrees of task feedback interval (ns for simple LD: $p = 0.099$). **d** Normalised training error integrated over learning (left) and dysmetria score at end of learning (right) of ccRNN with respect to cRNN for varying degrees of cerebral feedback horizons and task feedback intervals (left: simple LD task; right: online LD task). **\*\*** $p < 0.01$, **\*\*\*** $p < 0.001$, **\*\*\*\*** $p < 0.0001$ (two-sided paired *t*-test between cRNN and ccRNN). Error bars represent mean ± SEM across 10 different initial conditions. Source data are provided as a Source Data file.

training error (Fig. 4c). When the interval between external sensory feedback is increased, the benefits of the cerebellar-to-cerebral feedback predictions in the ccRNN model become clear. In contrast, for long feedback intervals, the feedback is too infrequent for either cRNN and ccRNN to be able to successfully learn the task. Next, we performed a detailed analysis of the co-dependency of the task (external) feedback interval and the cerebral feedback horizon (Fig. 4d). Our results show that ccRNN benefits learning and reduces dysmetria-like behaviours for intermediate feedback intervals provided that the cerebral feedback horizon is no longer than the task (external) feedback interval. This is a consequence of the cerebellum in our model being well placed to help the cerebrum learn when both internal and external feedback is not readily available.

**Similarity between cerebellar and cerebral feedback is task and learning dependent**

The cerebro-cerebellar facilitation of learning shown above depends on the ability of the cerebellum to provide the cerebral network with effective feedback predictions. To study the level of similarity between the predicted cerebral feedback and the theoretically optimal cerebral feedback as provided by gradient descent methods, we calculated the cosine similarity between cerebellar predictions and the optimal cerebral feedback in a range of tasks (Methods).

First, we measure the cosine similarity for tasks in which external sensory feedback is only provided at the end of the task – a variant of the simple LD task with feedback only at the end and the online visual

discrimination. This task setup allows for an easier interpretation of the similarity between cerebellar and cerebral feedback which should decay gradually from the end to the beginning of the task sequence. Indeed, we observe that the cerebellar-cerebral feedback similarity is higher closer to the point in which external sensory feedback is available (i.e. end of the task; Fig. 5a, b top; cf. Figs. 2, 3) and remains high over learning in particular for later points in the task (Fig. 5a, b bottom).

Next, we analyse the cosine similarity for conditions in which external feedback is available throughout the task. For this we consider the same visuomotor tasks as above (simple LD visuomotor, online LD visuomotor and online LD visuomotor). In these tasks, we observe more complex dependencies of the cerebro-cerebellar feedback similarity on task properties (Fig. 5c, d). For the simple LD task, we observe that the predictions made during earlier points in the task become more similar than those at later points throughout learning (Fig. 5c, d). These results suggest that the model tries to first learn to align later points in the task and then gradually attempts to learn to adjust earlier points. However, this is only possible in tasks such as the simple LD, which have regular feedback and can be fully learnt (i.e. achieve zero error). For the two remaining tasks, online LD and DD visuomotor tasks, and in contrast with the simple LD the similarity remains high throughout learning for later time points (Fig. 5d). This reflects the more challenging nature of these tasks and the need to continuously predict feedback as these tasks are never fully learnt (i.e. error remains higher than zero; cf. Fig. 3).

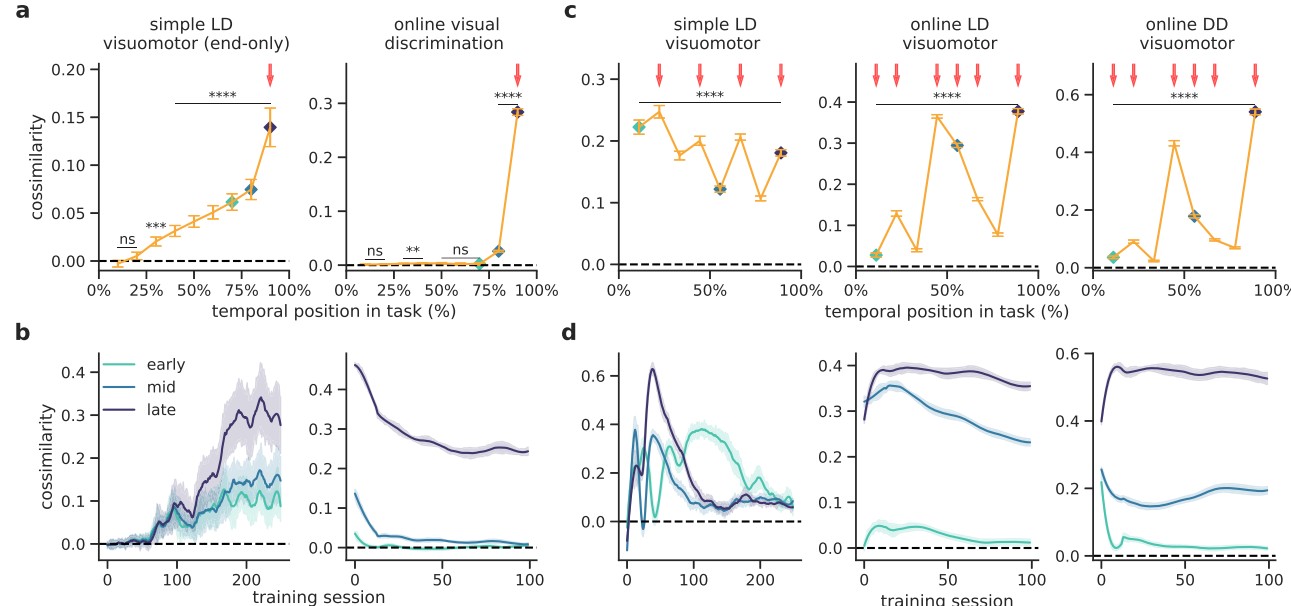

**Fig. 5 | Similarity between cerebellar and cerebral feedback is task and learning-dependent. a** Cerebro-cerebellar cosine similarity throughout task sequences that do not require intermediate external feedback: a simple line drawing with feedback only at the end of the task (LD end-only) and online visual discrimination (ns denotes not significant: simple LD visuomotor $p = 0.212$ (0%), $p = 0.520$ (25%); online LD visuomotor $p = 0.312$ (0%), $p = 0.06$ (25%), $p = 0.067$ (50%), $p = 0.386$ (60%). Here and in subsequent panels, red arrows indicate points in which external feedback is available. Cosine similarity throughout the tasks is calculated across all training sessions (see the "Methods" section). **b** Cerebro-cerebellar cosine similarity over learning for three-time points in the task: early (turquoise), mid (blue) and late (purple) in the task (cf. **a**). **c** Cerebro-cerebellar cosine similarity throughout the sequence for tasks with intermediate external feedback: simple line drawing (LD), online LD, online digitdrawing (DD). **d** Cerebro-cerebellar cosine similarity over learning for three different time points in the task (early, mid and late as in **b**). Dashed black line represents zero similarity. ** $p < 0.01$, *** $p < 0.001$, **** $p < 0.0001$ (two-sided paired $t$-test between cosine similarity and zero). Error bars represent mean ± SEM across 10 different initial conditions (20 for the simple LD visuomotor end-only task). Source data are provided as a Source Data file.

These results make predictions on when the cerebellum is able to better align with the cerebral feedback, which depends on task complexity, the properties of the task feedback, the exact task position and the learning stage. In particular, (i) for tasks with feedback only at the end (Fig. 5a), it predicts that cerebro-cerebellar feedback alignment should decay rapidly, and (ii) for tasks with regular external feedback (Fig. 5c) it predicts that cerebro-cerebellar feedback alignment should be stronger when more external feedback is provided.

## Learning shapes cerebro-cerebellar activity coupling

The cosine similarity results show that the cerebellar module learns to predict cerebral feedback. Because the cerebellum maps cerebral activity onto (predicted) cerebral feedback, this suggests changes in the coupling between cerebellar and cerebral neuronal representations throughout learning. To study the degree of cerebro-cerebellar coupling we calculate the pairwise correlations between neurons in the cerebral recurrent neural network and the neurons of the cerebellar network (see the "Methods" section). Although we observe a relatively small rise in the average cerebro-cerebellar coupling during the first few training sessions, as training progresses, there is a consistent decrease in the correlations (Fig. 6a).

To study more subtle changes in the correlation structure we use standard principal component analysis on the obtained pairwise correlations (Fig. 6b). The first principal component reflects the changes in the average cerebro-cerebellar coupling (Fig. 6b). The second principal component shows a delayed increase with respect to the first, followed by a sustained decrease in the cerebro-cerebellar coupling (see Fig. S14 for remaining components). These results are consistent with the need for the cerebellum to provide more effective feedback and thus be more coupled in the earlier learning phases. To study learning periods of consistent increases or decreases in coupling as training progresses we tracked the changes in correlations of cerebro-

cerebellar pairs in early, mid and late learning (Fig. 6c). We observe that early in learning — when most learning occurs — a large part of the population shows a consistent increase in correlations, but this rapidly changes as learning progresses with only a very small number of pairs showing increases in correlations later in learning.

To better assess the contribution of a plastic cerebellum to the cerebro-cerebellar coupling, we analysed a ccRNN in which the cerebellum does not learn. In this case we can still observe changes in cerebro-cerebellar coupling over learning for some tasks, which reflect changes in the RNN itself, but these are weaker when compared to the normal ccRNN (Fig. S15a). Cerebro-cerebellar correlations remain high throughout learning compared to a ccRNN with a plastic cerebellum. This is supported by their low-dimensional representations: whereas a plastic cerebellum leads to principal components that approach near-zero values after the initial learning phase (Figs. 6b, S14), in the case of the fixed cerebellum the principal components continue to fluctuate throughout learning (Fig. S15).

Although our model suggests a long-term decrease in the cerebro-cerebellar activity coupling, it highlights sub-populations that increase their coupling during specific periods of learning. This observation follows from our proposal in that the cerebellum is trained to map cerebral neuronal activity on cerebral feedback which depends on learning.

## Differential impact of cerebellar output and inferior olive on learning

In experimental neuroscience, a common paradigm is to inactivate the cerebellum in order to study its role in learning and behaviour. Here we perform in silico lesion experiments to reveal the impact of the modelled cerebellar feedback predictions during learning. First, we test cerebellar output lesions at different points during learning. In all tasks, we observe that inactivating the output of the cerebellar

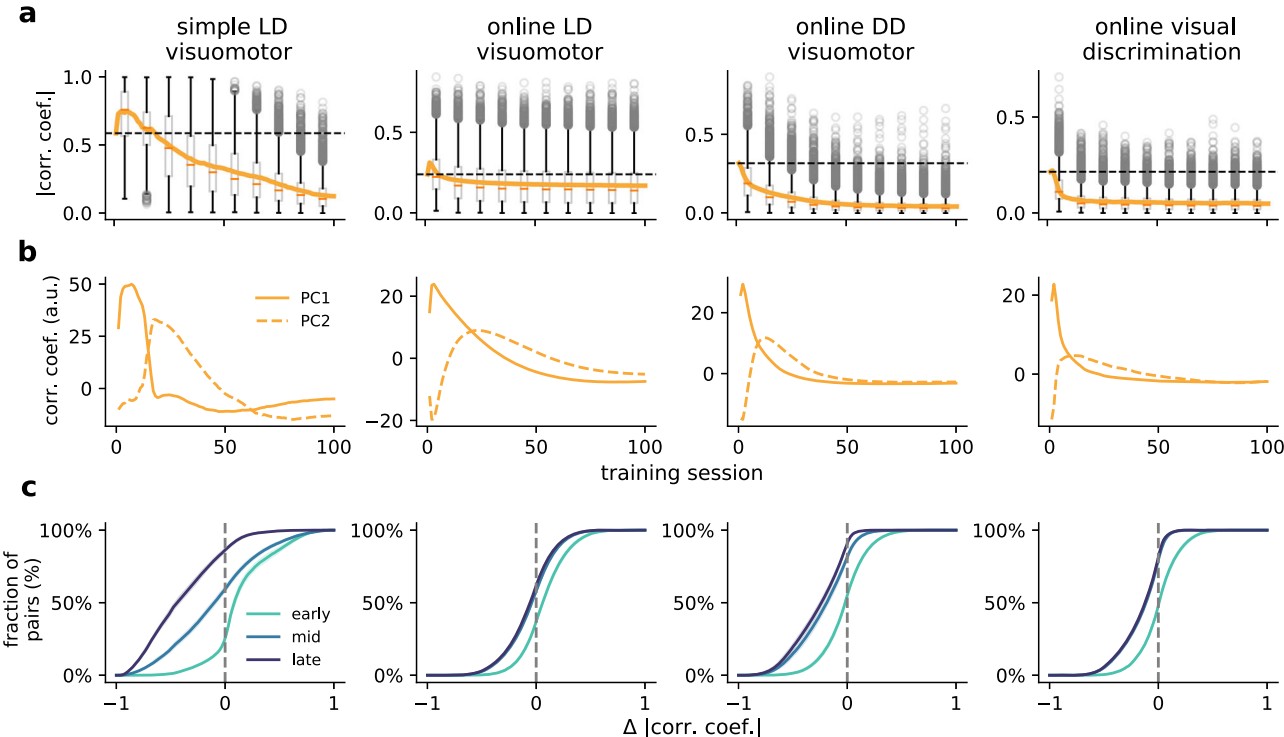

**Fig. 6 | Cerebro-cerebellar neuronal activity coupling over learning.**
**a** Distribution of pair-wise cerebro-cerebellar absolute correlation coefficients over learning for four tasks: simple LD, online LD, online DD and online visual discrimination. Orange line shows mean correlation coefficient. Boxplot shows median (horizontal dark orange line), interquartile range (IQR; box with centre at mean); whiskers show respective quartiles extended by 1.5 × IQR, where circles denote individual outliers beyond this range. Fully fixed ccRNN (i.e. without any form of plasticity in both networks) is given for reference (dashed line). **b** Change in

first two principal components of cerebro-cerebellar pair-wise correlation coefficients over learning (all components available in Fig. S14). **c** Cumulative plot of cerebro-cerebellar pairs with positive and negative changes in absolute correlation coefficients in early (session 1), mid (session 25) and late (session 80) learning. Data grouped across 10 different initial conditions, where for each condition we sample 600 active pairs for the simple LD visuomotor task and 1000 active pairs for the online tasks (see the "Methods" section). Source data are provided as a Source Data file.

module in early learning impairs further learning and performance (Fig. 7a, b). This is expected as the cerebellar network provides feedback predictions that facilitate cerebral learning. Interestingly, we observe that when the cerebellum is suddenly removed learning becomes worse than in the baseline model. This is likely due to the additional time taken to adapt to a new learning trajectory that no longer relies on cerebellar prediction. In contrast, cerebellar lesions performed later in learning do not have an impact on the simple LD visuomotor task, which is explained by the fact that for this task the model can achieve near-zero error, thus learning signals provided by the cerebellum are no longer needed. However, for all the online tasks we observe that inactivating the cerebellum even at later stages damages task performance. In these more realistic tasks, the cortical network still relies on the feedback provided by the cerebellum as it does not fully learn the task. Our results indicate that lesion studies should reveal a task-dependent nonlinear role of the cerebellum on cerebral learning.

Next, we assess the impact of disrupting cerebellar learning by modelling a complete lesion of our inferior olive-like error module (see the "Methods" section). This manipulation effectively stops cerebellar learning, thereby impacting the ability of the cerebellum to provide informative feedback learning signals to the cerebral network which may prevent the cerebral network from learning by perturbing its own learning trajectory. For all of the tasks that we model, inactivating cerebellar learning has a strong impact throughout training, making the model return to naive performance (Fig. 7c, d). Thus, simulated "inferior olive" lesions predict that if the cerebellum cannot learn it would result in a stronger negative impact on task learning than ablating the cerebellum itself, in line with recent experimental observations[50]. This further suggests that it is

critical for the cerebellum to learn rapidly to be able to provide informative predictions.

## Cerebro-cerebellar model facilitates learning in a visual-language task
Our framework does not only apply to sensorimotor tasks but should generalise to virtually any task within the grasp of current neural network models. To test the generability of our model and inspired by language tasks in which cerebellar patients have shown deficits[14,51–54] we test our models in a caption generation task which models the recreating sentence task studied by Guell et al.[14]. In this task the network needs to generate a textual description for a given image, similar to the task conducted by Guell et al.[14]. All models have two components: a pretrained convolutional neural network (CNN) that extracts a lower dimensional representation of the image, and a cRNN or ccRNN on top which is trained to map the low dimensional visual input to captions that describe the image (Fig. 8a).

We use a standard machine learning dataset[55] and the networks are trained to predict the next word (see the "Methods" section). We find that ccRNN models can exhibit faster learning (Fig. 8b) and better generalisation[56] (Fig. S16) when in the presence of short cerebral feedback horizons (≤40%, Fig. 8d). All models produce reasonable captions for images unseen during training, but ccRNN models tend to produce captions that better capture the context and semantics of the task (Figs. 8c, S17), consistent with the poorer descriptions of images generated by cerebellar patients[14]. In our model, the ability to generate more accurate textual descriptions of images is due to the ability of the ccRNN model to perform better temporal credit assignment by providing feedback estimates beyond the cortical feedback horizon.

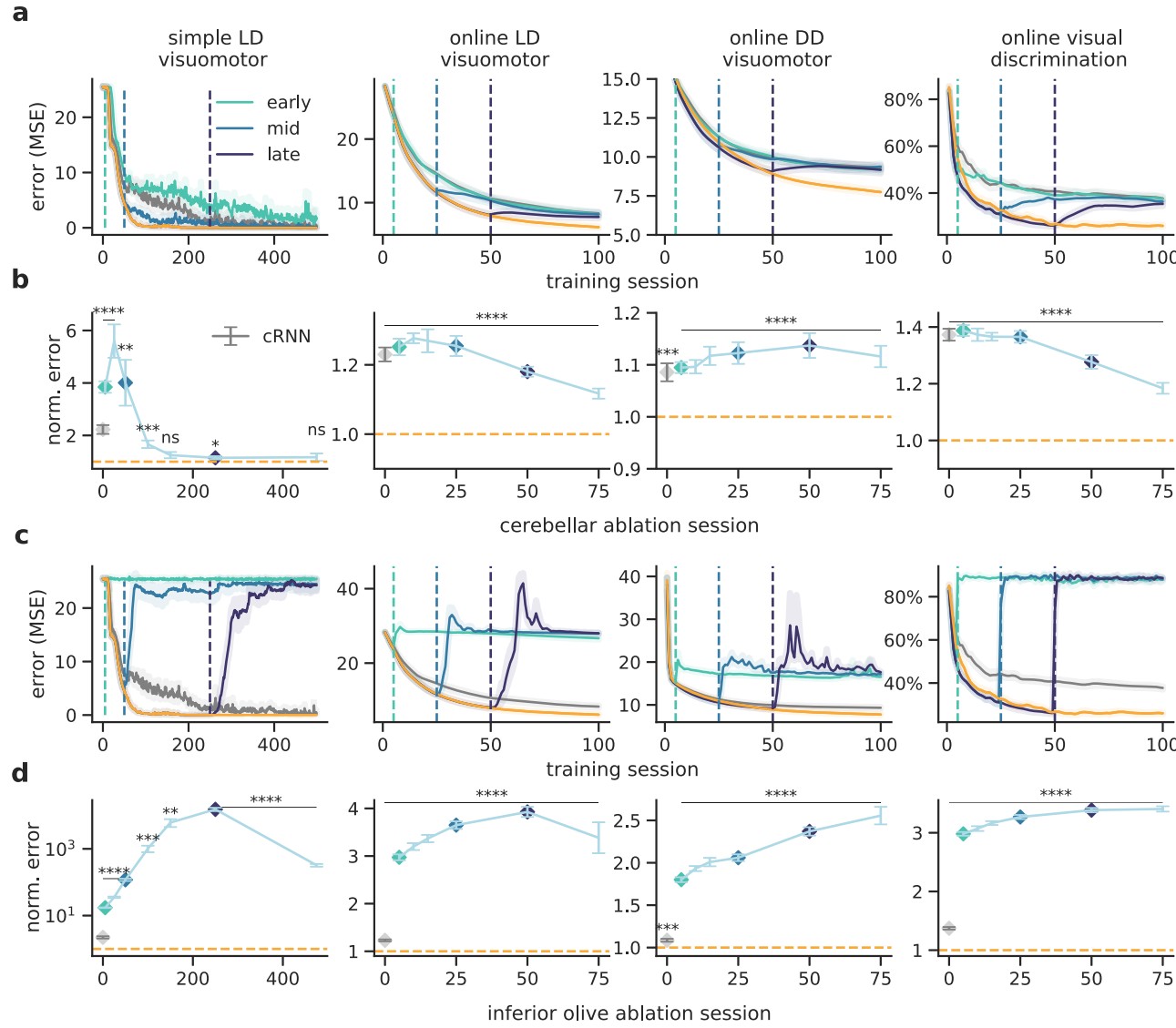

**Fig. 7 | Inactivating cerebellar output and inferior olive have a differential impact on learning. a** Complete cerebellar lesion at different points during learning. Vertical lines represent at which point during training the cerebellum was inactivated in the ccRNN model. In gray and orange are shown the baseline performances of the cerebral RNN and ccRNN, respectively. **b** Normalised error after cerebellar lesion throughout learning with respect to ccRNN (ns denotes not significant: simple LD visuomotor $p = 0.062$ (session 150), $p = 0.162$ (session 475)). Gray denotes normalised error for cRNN. **c** Complete inferior-olive lesion at different points during learning. Vertical lines represent point of lesion of the ccRNN model. In gray and orange are shown the baseline performances of the cerebral RNN and ccRNN, respectively. **d** Normalised error after inferior-olive lesion throughout learning with respect to ccRNN. Gray denotes normalised error for cRNN. *$p < 0.05$, **$p < 0.01$, ***$p < 0.001$, ****$p < 0.0001$ (two-sided paired $t$-test between ccRNN (ablation) and ccRNN (control)). Error bars represent mean ± SEM across 10 different initial conditions. Source data are provided as a Source Data file.

Finally, we use a language metric (SPICE[57]) to measure the quality of the generated captions. These results show that the ccRNN generates richer captions and that it is particularly beneficial for longer captions (Fig. 8e). This suggests that ccRNN is able to learn richer visuo-language contextual information.

## Discussion

Inspired by recent deep learning developments, here we have introduced a systems-level computational model in which cerebellar networks predict cerebral feedback (Fig. 1). In this scheme cerebro-cerebellar loops decouple cerebral cortical networks from future feedback signals. We show that the ccRNN model accelerates learning and improves task behaviour in a range of sensorimotor and cognitive tasks (Figs. 2, 3 and 8). Our results are consistent with observed motor and cognitive deficits in cerebellar patients. Our model makes a number of predictions in terms of (1) task properties (Figs. 4 and 5), (2)

cerebro-cerebellar representations and coupling (Figs. 2 and 6), and (3) the differential role of the cerebellum and the inferior olive throughout learning (Fig. 7).

Experimental studies have shown that incomplete or delayed external sensory feedback is important for learning[46,58,59]. Our model proposes that the cerebellum plays an important role in facilitating motor learning when in the presence of incomplete or delayed feedback. Furthermore, our work suggests that cerebro-cerebellar networks are ideally placed to facilitate learning when task feedback is presented intermittently, at medium frequencies with respect to the task sequence. Similarly, our results suggest that cerebellum-dependent dysmetria should be more prevalent for tasks with intermediate to long inter-feedback intervals. Although there is a wide range of studies investigating the role of external sensory feedback in learning[58,60] and the precise timing of feedback is known to be important for cerebellar function[10,61], it remains to be tested what are the optimal properties of

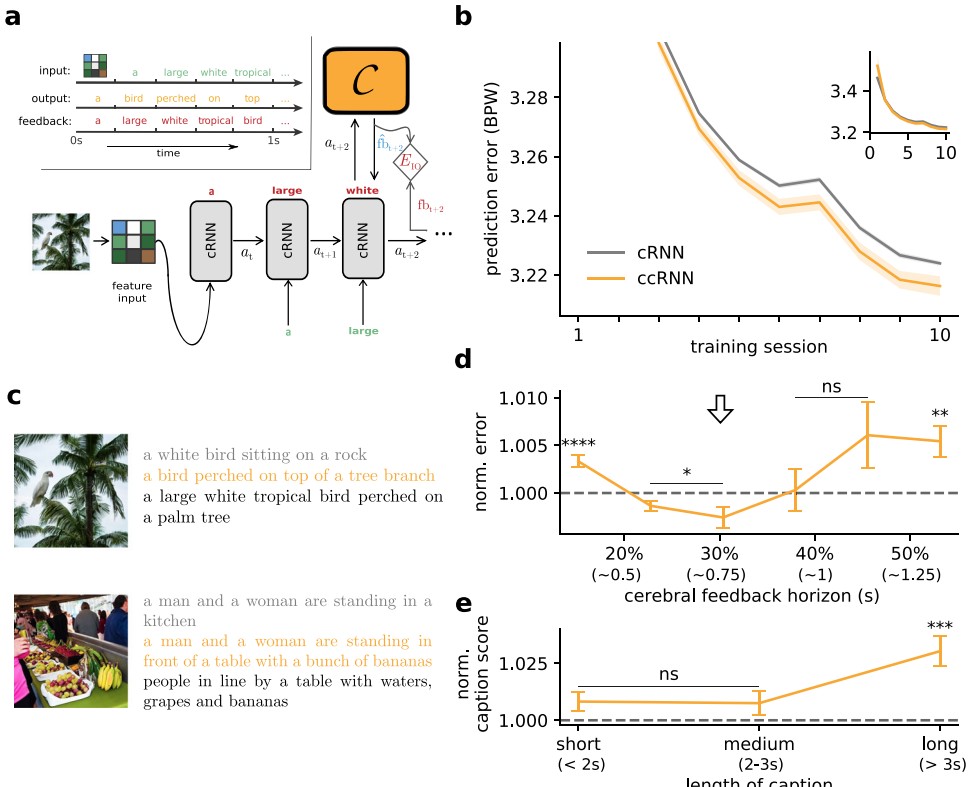

**Fig. 8 | Cerebro-cerebellar model facilitates learning in a visual-language task.** **a** Schematic of the model used in a visual-language task. The image is first processed by a (pretrained) convolutional neural network modelling the visual cortex. The resulting feature vector is then provided to the cerebral RNN which is trained to predict the next word given the previous words of a provided "gold standard" image caption. The cerebellum module $\mathcal{C}$ is only applied to the cRNN. Top left: task structure with example input image and words (green), ccRNN output words (orange) and target caption (red). **b** Learning curves in bits per word (BPW), lower values indicate better understanding of the language on validation set for cerebral feedback horizon of four timesteps (inset shows complete learning curve). **c** Two example images from the validation set with corresponding model captions and gold standard captions (black). The images shown here were generated on dee-pAI.org for illustration purposes only. **d** Normalised model performance across different degrees of feedback horizon in the cerebral network (ns denotes not significant: $p = 0.891$ (40%), $p = 0.116$ (45%)). **e** Normalised caption score (see the "Methods" section) as a function of caption length (ns: $p = 0.075$ (short), $p = 0.189$ (medium)). *$p < 0.05$, **$p < 0.01$, ***$p < 0.001$, ****$p < 0.0001$ (two-sided paired $t$-test between cRNN and ccRNN). Error bars represent mean ± SEM across 10 different initial conditions. Source data are provided as a Source Data file.

task feedback for learning. Taken together, we suggest cerebellar-mediated feedback predictions to be particularly important for temporally challenging tasks with sparse feedback.

Our representational analyses demonstrate that the cerebellum develops task-specific representations. Recent fMRI studies have observed that different regions of the cerebellum encode task-specific representations for different task domains[23,62]. Similarly, our model predicts the need for different cerebellar modules to provide feedback estimations to the cerebral cortex for specific task domains. We have also studied the level of coupling between cerebellar and cerebral neural activity. Our results demonstrate an initial rise in correlations which coincides with steep periods of learning followed by a general decay in the coupling during the remaining periods of learning. This general decay in coupling is also reflected in our simulated cerebellar lesions which echo the existing literature in that after a task is consolidated in the cerebrum it becomes less cerebellar-dependent[63,64].

In line with previous theoretical accounts[6,7,9] we suggest that the cerebellar error function is computed by the *inferior olive*, which drives learning in the cerebellum via the climbing fibres. This cerebellar error function is a combination of true sensory feedback and self-predicted (bootstrapped) error signals (Fig. 1b), which is analogous to the bootstrapping principles commonly used in reinforcement learning[65]. The use of self-predictions in the cerebellum suggests the existence of different forms of feedback to the inferior olive from potentially multiple cerebellar modules[66], consistent with

cerebellar-inferior olive connectivity[67]. Moreover, when ablating the inferior olive we show that task performance becomes severely impaired. This is due to the cerebellum being unable to learn, thereby providing irrelevant feedback signals back to the cerebral cortex. These results suggest non-trivial consequences of lesions for cerebro-cerebellar interactions.

While our model is consistent with experimental observations, there are several biological features that we have not considered. In particular, experimental studies suggest that the cerebellum can influence cerebral learning processes via its projections via the thalamus[68–71]. This is in line with ccRNN in which the cerebellum predicts feedback signals that contribute directly to cerebral learning. However, we have assumed direct long-range projections with the cerebral cortex whereas in biology these projections are mediated through the thalamus and pons. It is possible that both structures may provide bottlenecks that filter out non-relevant information, such as poor estimated feedback (Figs. 2d, 3d) that would impair cerebral learning. In addition, cerebellar-thalamic-cerebral projections are known to target distal dendrites of pyramidal cells[72,73], which have been proposed to encode feedback error signals by biologically-plausible deep learning models[74,75]. These dendritic-encoded error signals are akin to the gradient descent errors that we use to model cortical feedback signals. In future work, it would be of interest to combine our work with biological gradient descent models.

Throughout this paper, we have assumed the existence of cerebral prediction error modules, which compare the output of a given cerebral

area with the desired task output to generate a feedback teaching signal for the cerebral cortex. There is evidence of prediction errors across different brain areas, for example, sensorimotor prediction errors in the neocortex[76,77] or reward prediction errors in the VTA[1,78]. For simplicity, here we have focused on supervised (Figs. 2, 3) and unsupervised (Fig. 8) prediction errors, but these can in principle be readily replaced by reward-based prediction errors[1,79]. This would predict reward-specific encodings in the cerebellum as observed recently[80–82]. Indeed, our model is of particular relevance to reinforcement learning due to the prevalence of sparse and delayed rewards (Fig. 4).

Finally, our model shares common features with classical internal models of the cerebellum[6,7] (Table S1). In the forward model of sensorimotor control, the cerebellum receives an efferent copy of the motor commands and the respective external sensory feedback[8,83]. With these two input streams, the forward model learns to predict the sensory consequences of motor commands. We and others have argued that a similar predictive model can in principle be applied to higher-order brain regions such as the prefrontal cortex and the temporo-parietal cortex which are involved in the planning of cognitive behaviour and decision-making[16,17,24,26] (Fig. 1a). In line with forward models, the cerebellar module of ccRNN receives an efferent copy of the cerebral neural activity and cerebral feedback. Given these signals, the cerebellum learns to predict future cerebral feedback.

Overall, our work offers a theoretical framework with which to study cerebro-cerebellar interactions, being consistent with experimental observations while making a large number of testable predictions across multiple levels of interrogation.

## Methods

In all our experiments we model a cerebral area $A$ as a long short-term memory recurrent neural network (LSTM)[84] with parameters $\theta$ which has recently been mapped onto cortical microcircuits[85]. A (trained) linear readout is attached to the LSTM output states which provides the final model output to a supervised error module $E^{\text{task}}$, which below we refer to as $E$.

In the cerebro-cerebellar RNN model (ccRNN) we attach a feedforward cerebellar module $C$ with independent parameters $\Psi$ to the RNN with reciprocal connections (Fig. 1). The cerebellar module is equivalent to the "synthesiser" as used by Jaderberg et al.[33] in the backward case. That is, the cerebellar module receives a copy of the RNN activity $\mathbf{a}_t$ (both cell and output LSTM states) and sends back a prediction of the future feedback (or error gradients) with respect to that activity, $C(\mathbf{a}_t)$.

To generate the desired cerebral temporal feedback (error gradients) we use backpropagation through time (BPTT)[86]. To highlight the link between BPTT-derived feedback and the cerebellar predicted feedback we start out from first principles closely following Jaderberg et al.[33,87]. BPTT is the standard solution to generate feedback with respect to parameters $\theta$ in artificial recurrent neural networks. Suppose that the task is of length $T$, full BPTT considers all error signals from current time $t$ up to the end of the task $T$, $\sum_{t'=t}^{T} E_{t'}$, and defines parameter updates as $\theta \leftarrow \theta - \alpha \Delta\theta$ where $\Delta\theta = \sum_{t'=t}^{T} \frac{\partial E_{t'}}{\partial \theta}$. Note that the summation goes forward in time, but for each $t'$ with external feedback there is a BPTT feedback that is calculated backward in time until $t$ (Eq. 1). However, doing full BPTT means that we have to store the entire sequence of computational steps to then generate feedback with respect to a given point in time $t$. Instead, in practice, BPTT over a limited time horizon (or truncation) $K$ − known as *truncated* BPTT − is commonly used (Fig. 1):

$$\sum_{t'=t}^{T} \frac{\partial E_{t'}}{\partial \theta} \approx \left( \sum_{t'=t}^{K} \frac{\partial E_{t'}}{\partial \mathbf{a}_t} \right) \frac{\partial \mathbf{a}_t}{\partial \theta} = \mathbf{fb}_{t'} \frac{\partial \mathbf{a}_t}{\partial \theta} \tag{1}$$

where $\mathbf{fb}_{t'}$ designates the gradient information derived from the $K$-horizon BPTT and which we refer to as *cerebral feedback* (see dashed red lines in Fig. 1b). This is appealing both from a computational and biological perspective as it prevents long sequences of BPTT. Indeed, one might consider the imposed horizon $K$ as a constraint of the nervous system in retaining temporal information.

However, by enforcing a temporal horizon of $K$ steps RNNs lose the ability to perform long temporal credit assignments. In our model, the cerebellum recovers the ability to perform longer temporal credit assignments. In particular, the cerebellum in the ccRNN is tasked with predicting future feedback/gradients giving the current state of the cerebral RNN (Eq. 1), thereby providing feedback estimates to the RNN that go beyond the $K$-horizon. The final error gradient used by the RNN is then a combination of cerebral and cerebellar feedback as follows:

$$\sum_{t'=t}^{T} \frac{\partial E_{t'}}{\partial \theta} = \left( \sum_{t'=t}^{K} \frac{\partial E_{t'}}{\partial \mathbf{a}_t} \right) \frac{\partial \mathbf{a}_t}{\partial \theta} + \left( \sum_{t'=K+1}^{T} \frac{\partial E_{t'}}{\partial \mathbf{a}_K} \right) \frac{\partial \mathbf{a}_K}{\partial \theta}$$

$$\sum_{t'=t}^{T} \frac{\partial E_{t'}}{\partial \theta} \approx \left( \mathbf{fb}_{t'} + \hat{\mathbf{fb}}_{t'>K} \frac{\partial \mathbf{a}_K}{\partial \mathbf{a}_t} \right) \frac{\partial \mathbf{a}_t}{\partial \theta} \tag{2}$$

$$\Delta\theta \approx \left( \underbrace{\mathbf{fb}_t}_{\text{cerebral feedback}} + \underbrace{C(\mathbf{a}_K)}_{\text{cerebellar feedback}} \frac{\partial \mathbf{a}_K}{\partial \mathbf{a}_t} \right) \frac{\partial \mathbf{a}_t}{\partial \theta}$$

where $C(\mathbf{a}_K)$ denotes the cerebellar predictions of *future feedback* beyond horizon $K$ and $\frac{\partial \mathbf{a}_K}{\partial \mathbf{a}_t}$ represents the changes in cerebral activity over time. Note that the cerebellum predicts future feedback with respect to the RNN activity at the end of cerebral feedback horizon (i.e. $\mathbf{a}_K$). Also note that if we set $C(\mathbf{a}_K) = 0$ then we simply have the standard truncated-BPTT formulation (Eq. 1). The parameters updates can then, in principle, be performed at the beginning of each horizon (i.e. $K$ time step) or accumulated with the other updates and performed jointly at the end (see more on this point in the Experimental Details below).

A key consequence of the cerebellum predicting future feedback is that *strong* or *long* feedback signals (i.e. $T \gg 0$) are no longer necessary, thus decoupling learning in the cerebral network from future feedback signals. For this reason, we focus on weak forms of BPTT with relatively small temporal horizons, in which we model only $K$ time steps of feedback into the past from an error signal $E$ (truncated BPTT as defined above). In our experiments the size of $K$ − which we report as a percentage of the task length (cerebral temporal horizon) − varies but is generally small. For example, for the simple line drawing task we used a one-step BPTT (i.e. $K = 1$; Fig. 2). Note that in the main text as we describe a simpler case of $K = 1$ (as used in the simple line drawing task) we use $C(\mathbf{a}_t)$ to refer to the cerebellar feedback prediction from the end of the current horizon, i.e. $C(\mathbf{a}_t) = C(\mathbf{a}_K) \frac{\partial \mathbf{a}_K}{\partial \mathbf{a}_t}$.

### Cerebellar learning

The cerebellar parameters $\Psi$ are themselves learnt but to optimise a distinct, specialised error $E^{\text{IO}}$ which we posit to be computed at the *inferior olive*, the classical teacher of the cerebellum[6,7]. This is defined by the difference between cerebellar output and a target feedback signal $\overline{\mathbf{fb}}_t$, i.e. $E_t^{\text{IO}} = ||C(\mathbf{a}_t) - \overline{\mathbf{fb}}_t||$. Similar to the cerebral network we update cerebellar parameters using gradient descent: $\Psi \leftarrow \Psi - \alpha^{IO} \Delta\Psi$, where $\Delta\Psi = \frac{\partial E^{\text{IO}}}{\partial \Psi}$.

Ideally, we would simply set the target feedback as the true (desired) cerebral feedback. However, this would require an arbitrarily long number of steps of true cerebellar feedback, exactly what we propose that is not required with a cerebellar network. How should the cerebellum itself learn about future feedback? One elegant solution, which we take from Jaderberg et al.[33], is to combine the currently available error with bootstrapped future cerebellar predictions (i.e. self-predictions). Formally, using the same notation as

Eq. 2, the trained target for $C(\mathbf{a}_T)$ is

$$\overline{\mathbf{fb}}_T = \frac{\partial E_{\leq 2T}}{\partial \mathbf{a}_T} + C(\mathbf{a}_{2T})\frac{\partial \mathbf{a}_{2T}}{\partial \mathbf{a}_T} \qquad (3)$$

Note the resemblance of Eq. 3 to Eq. 2: in each case, we consider a mixture of nearby "cerebral" error signals beyond which we rely on cerebellar prediction. It is also useful to compare Eq. 3 with standard reinforcement learning rules (e.g. temporal difference learning algorithm) which rely on similar bootstrapping principles[65]. We verify the importance of the bootstrap component in the online visual discrimination task (Fig. S13).

## Continuous and discrete time versions of RNN

To infer approximate timescales of the RNN we follow the approach of Song et al.[88] in considering its continuous-time dynamics. In particular, we can express a general continuous time version of LSTM dynamics as

$$\mathbf{f} = \sigma\left(\theta_f^{in}\mathbf{x} + \theta_f^{rec}\mathbf{a} + \mathbf{b}_f\right) \qquad (4)$$

$$\mathbf{i} = \sigma\left(\theta_i^{in}\mathbf{x} + \theta_i^{rec}\mathbf{a} + \mathbf{b}_i\right) \qquad (5)$$

$$\mathbf{o} = \sigma\left(\theta_o^{in}\mathbf{x} + \theta_o^{rec}\mathbf{a} + \mathbf{b}_o\right) \qquad (6)$$

$$\tilde{\mathbf{c}} = \tanh\left(\theta_c^{in}\mathbf{x}_t + \theta_c^{rec}\mathbf{a} + \mathbf{b}_c\right) \qquad (7)$$

$$\tau\dot{\mathbf{c}} = -(1-\mathbf{f})\circ\mathbf{c} + \mathbf{i}\circ\tilde{\mathbf{c}} \qquad (8)$$

$$\mathbf{a} = \mathbf{o}\circ\mathbf{c} \qquad (9)$$

where $\mathbf{f}$, $\mathbf{i}$, $\mathbf{o}$ denote the LSTM forget, input, and output gates, respectively, $\tilde{\mathbf{c}}$, $\mathbf{c}$, $\mathbf{a}$ denote the candidate cell state, cell state, observable state, respectively. $\sigma$ denotes the logistic function $\sigma(x) = \frac{1}{1+\exp(-x)}$. We use $\dot{\mathbf{c}} = \frac{\partial\mathbf{c}}{\partial t}$ to denote the derivative of the cell state with respect to time which is scaled by the neuronal time constant $\tau$. We set $\tau = 100$ ms in line with previous RNN-based models[88,89] for all tasks except the more cognitive image captioning task for which we assume a slower time constant $\tau = 200$ ms. The weight and bias vectors $\theta^{in}$, $\theta^{rec}$, $\mathbf{b}$ are to be learned during training.

Applying a first-order Euler approximation on equations with a time-discretization step $\Delta t$ then yields

$$\mathbf{f}_t = \sigma\left(\theta_f^{in}\mathbf{x}_t + \theta_f^{rec}\mathbf{a}_{t-1} + \mathbf{b}_f\right) \qquad (10)$$

$$\mathbf{i}_t = \sigma\left(\theta_i^{in}\mathbf{x}_t + \theta_i^{rec}\mathbf{a}_{t-1} + \mathbf{b}_i\right) \qquad (11)$$

$$\mathbf{o}_t = \sigma\left(\theta_o^{in}\mathbf{x}_t + \theta_o^{rec}\mathbf{a}_{t-1} + \mathbf{b}_o\right) \qquad (12)$$

$$\tilde{\mathbf{c}}_t = \tanh\left(\theta_c^{in}\mathbf{x}_t + \theta_c^{rec}\mathbf{a}_{t-1} + \mathbf{b}_c\right) \qquad (13)$$

$$\tau\mathbf{c}_t = (1-\alpha)\mathbf{c}_{t-1} + \alpha\mathbf{f}\circ\mathbf{c}_{t-1} + \alpha\mathbf{i}\circ\tilde{\mathbf{c}}_t \qquad (14)$$

$$\mathbf{a}_t = \mathbf{o}_t\circ\mathbf{c}_t \qquad (15)$$

where $\alpha = \frac{\Delta t}{\tau}$. In our experiments, we use $\alpha = 1$ which recovers the standard dynamics of the discrete LSTM[84]. In this case, the length of each timestep is the same as the neuronal time constant, i.e. $\Delta t = \tau$.

## Other biological mappings of our framework

Here we describe other possible mappings between the proposed framework (cerebellum as a decoupling machine) and forward and feedback processing in the cerebral cortex.

**Cerebellum as a spatial feedback decoupler.** Our paper focuses on temporal problems being solved by a cerebral area modelled as a recurrent neural network (RNN) to which a cerebellar network provides predictions of future errors/feedback with respect to that area. An analogous biologically relevant system also arises, however, when one considers cerebral processing in space using feedforward computations involving several distinct regions (Fig. S1).

This setup – where the "main" (cerebral) network is a feedforward composition of multiple brain regions – was also considered in Jaderberg et al.[33]. Now, as opposed to predicting errors that occur strictly at later points in time, the role of the cerebellar network is to predict errors which occur in later brain regions. The result is that an earlier region has access to its feedback (predicted by the cerebellum) without the need to wait for the later forward/backpropagation of spatial activity. Formally, if (with abuse of notation) we assume cerebral processing as a sequence $\{a_i\}_{i=1}^N$ of feedforward computations: $A(\mathbf{x}) = (a_N \circ a_{N-1} \circ \cdots \circ a_1)(\mathbf{x})$ which defines a final error function $E(A(\mathbf{x}))$, then the cerebellar network can provide predicted feedback at a given brain area as soon as its activities are computed: $C(\mathbf{a}_i) := \hat{\mathbf{fb}}_i = \frac{\partial\hat{E}}{\partial\mathbf{a}_i} \approx \frac{\partial E}{\partial\mathbf{a}_i}$.

This perspective could effectively speed-up feedback processing across the brain. This interpretation of the model is consistent with cerebellar-thalamo-cerebral projections targeting distal dendrites, which have been proposed as the site of error or feedback encoding which underlies efficient hierarchical learning[74,75].

**Cerebellum as a forward decoupler.** In classical cerebellar theory, the complement to the forward model hypothesis is the inverse model, in which the cerebellum predicts motor commands[5], or even implicit mental predictions to solve a problem[24], directly. Again we can consider this under the proposed framework, but now using its *forward* prediction version.

In this case, the role of the cerebellum is not to predict future feedback activity, but the feedforward activity itself, i.e., $C(\mathbf{a}_i) = \hat{\mathbf{a}}_j$ for some later region $j > i$. $\hat{\mathbf{a}}_j$ is fed as a replacement to region $j$, making it forward decoupled from a potentially slower intermediate processing $\mathbf{a}_j \circ \mathbf{a}_{j-1} \circ \cdots \circ \mathbf{a}_{i+1}$.

Functionally this would provide the organism with fast inputs (e.g. motor commands or potential mental solutions) without the need for potentially slower cerebral processing (Fig. S1b). We also point out the relevance of direct predictions of later activity in the temporal case, where the cerebellum strictly predicts motor activity at later time-steps, as suggested in ref. 90. A broad comparison between this framework and the cerebellar internal model hypothesis is shown in Table S1.

## Experimental details

To reduce learning instability we scale the cerebellar predicted feedback (Eq. 2) by 0.1[33]. Both cerebral and cerebellar parameters are optimised using the feedback described above together with ADAM for overall learning efficiency[91]. Training the model involves iterating over training sessions for a given dataset, which is split into batches. During training, model parameter gradients are accumulated over the truncations within the batch (as defined by the cerebral feedback horizon) and the parameters are updated at the end of the batch. Note that for ccRNN these updates could in principle take place after each individual truncation since the error gradient will always at least contain the cerebellar prediction. However, because ADAM increases the learning rate in the presence of small gradients, as is the case of cerebellar-derived gradients, updating at every truncation would make the ccRNN−cRNN comparison unfair. We conducted tests in which we

updated parameters at every truncation and we get qualitatively similar results.

In each experiment, all initial RNN parameters are drawn from a uniform distribution $\mathcal{U}(-\frac{1}{\sqrt{n_{RNN}}}, \frac{1}{\sqrt{n_{RNN}}})$, where $n_{RNN}$ is the number of RNN units. The weights of the readout network and the feedforward weights of the cerebellar network (other than the final layer) are initialised according to $\mathcal{U}(-b_k, b_k)$ where $b_k$ denotes the "kaiming bound" He et al.[92] (slope $s = \sqrt{5}$), and the biases are drawn from $\mathcal{U}(-\frac{1}{\sqrt{n_{in}}}, \frac{1}{\sqrt{n_{in}}})$, where $n_{in}$ denotes the input size of the layer. The last layer (both weights and bias) of the cerebellar network is zero-initialised, so that the estimated feedback at the start is zero[33]. This initialisation makes learning overall more stable but does not change our results qualitatively. To demonstrate this we do not zero-initialise the cerebellar output in one of the tasks (simple visuomotor task).

During learning, we employ truncated BPTT as follows. Given an input sequence of $N$ timesteps $\mathbf{x}_1, \mathbf{x}_2, ..., \mathbf{x}_N$ and a temporal horizon $K$, we divide the sequence into $K$ sized truncations. In other words, the sequence is now made up of truncations of $(\mathbf{x}_1, ..., \mathbf{x}_K), ..., (\mathbf{x}_{(m-1)K+1}, ..., \mathbf{x}_{mK}), (\mathbf{x}_{N-r}, ..., \mathbf{x}_N)$, where $N = mT + r$ for positive integers $m, r$ with $0 \le r < K$. Note that, along with the value $K$, how well the sequence is divided into truncations (i.e. values $m, r$) can itself influence learning (e.g. Fig. 3d).

In the all visuomotor tasks, to test the effect of predicted feedback against the availability of task feedback signals which occur at any timestep where an external teaching signal is provided, we vary the *external feedback interval*. Given feedback interval $n$, the target is only available every $n$ timesteps. This is analogous to the rate at which one receives sensory information whilst performing a task (e.g. drawing freehand).

The error with respect to these (potentially sparse) available targets is reported as the training error in the main text, or simply error. For the drawing tasks, we also consider the total error with targets at every timestep, whether available during training or not. This quantifies the "smoothness' of the model output (i.e. the straightness of the line between two available targets). We refer to this metric at the end of training as the *dysmetria score*. For the visuomotor discrimination task we define the dysmetria score as 1 minus the probability of the model's most likely choice, which quantifies the model uncertainty.

Hyperparameters are standard and were selected based on multiple trial runs.

*Delta and normalised error*: To calculate the delta and normalised error with respect to a given model we take the difference or ratio of total errors during learning (all training sessions). For example, the normalised error of ccRNN with respect to cRNN is $\frac{error(ccRNN)}{error(cRNN)}$. Note that in the ablation case we compare against a "healthy" ccRNN and only consider the respective errors post-ablation. e.g. the normalised error for a model with cerebellar ablation at session 50 is $\frac{error(ablated)_{>50}}{error(ccRNN)_{>50}}$.

*Cerebro-cerebellar coupling*: To analyse how the coupling between the cerebral and cerebellar networks changes over learning we consider the (absolute) Pearson correlation between a given cerebral (LSTM) unit and a given unit in the cerebellar hidden (granular) layer over different bins during training. Values given are the average correlation over all RNN/cerebellar unit pairs. The PCA analysis is performed on the time × cerebro-cerebellar pairwise correlation coefficient matrix.

We found that towards the end of learning, several units in the cerebellar hidden layer became silent. This led to undefined pairwise correlations for those units. For this reason, we sampled pairs of units that were active (non-zero) throughout training. For the simple LD visuomotor task, we sampled 600 pairs for each initial condition (6000 pairs in total); for the online tasks, we sampled 1000 pairs for each initial condition (10,000 pairs in total).

*Computing details*: All experiments were conducted on the Blue-Pebble supercomputer at the university of Bristol; mostly on GPUs

(GeForce RTX 2080 Ti) and some on CPUs. We estimate the total compute time (including unreported results) to be in the order of ~2000 h.

**Simple line drawing visuomotor task.** In the simple line drawing task, an LSTM network receives a discrete input cue that signals the network to either (1) do not move or (2) draw a line in 2D space over a period of 10 timesteps. Here we set 6 distinct non-zero input-target pairs $\{(\mathbf{x}_i, \mathbf{y}_i)\}_{i=1}^6$, where each input $\mathbf{x}_i$ is a (one dimensional) integer $\in \{\pm 1, \pm 2, \pm 3\}$, and the targets $\{\mathbf{y}_i\}_{i=1}^6$ are lines whose endpoints lie equidistantly on a circle centred on the origin with radius 10. To make the task more realistic we also consider a 7th target in which the network must remain quiet at the centre of the drawing screen, modelling periods in which the animal is not actively performing the task. Once an input cue is received at timestep $t_0$, the model receives no new information (i.e. all future input is set to zero). The model is trained to minimise the mean squared error (MSE) between its output and the cue-based target. External sensory feedback is presented as the target line sampled every other time step starting from the first time step.

The cerebral network is modelled by one hidden layer of 50 LSTM units and the cerebellar network by one hidden layer of 400 neurons. The learning rate is set to 0.001. Each epoch comprises 16 batches with 50 randomised examples. Unless explicitly stated we use a truncation size of $K = 1$ which covers 10% of the total task duration. Model results are averaged over 10 random seeds (with error bars), where each seed determines the initial parameters of the network.

**Online visuomotor tasks.** For each online visuomotor task (Fig. 3) we use a standard dataset of handwritten digits (MNIST dataset). Unlike the simple line drawing task, the model now receives a temporal stream of input. In particular, given a $28 \times 28$ handwritten (MNIST) digit, the input at timestep $t$ is a vector of pixel values at the row $t$ model of the image (see Fig. 3a, right). The input is thus of dimension 28 and is presented over a total of 28 timesteps.

In each case, we have one hidden layer of 30 LSTM units in the main model and one hidden layer of 300 hidden units in the feedforward cerebellar network. Data was presented in batches of 50 with a learning rate of 0.0001.

Training and validation data were assigned a 4:1 split, containing 48,000 and 12,000 distinct image/number pairs, respectively. Unless explicitly stated, the truncation value was $K = 3$ which is ~10% of the task duration. Model results are presented over 10 random seeds.

**Online line drawing visuomotor task.** In this variant, each number 0–9 MNIST image is allocated an associated $xy$ position on the edge of a circle centred at 0 with radius 10, and during the presentation of the input must draw a line of equally spaced points towards that position (Fig. 3a, left). With the model output being a vector of size 2, the training loss is defined at the end by the mean squared error (MSE) between the output of the model and the points forming the target line.

**Online digit drawing visuomotor task.** Like the online line drawing task, in this variant, the model outputs a sequence of 2D coordinates during input presentation. The target sequence however is now highly non-linear, and in this case is a template of the number represented by the MNIST image (Fig. 3a, middle). The model is then trained to minimise the MSE between the model output and that target shape.

For each digit, the corresponding target drawing lies in $[0, 1] \times [0, 1]$, such that the gap between each successive point is equivalent. All model drawings begin in the top left corner (except for digit 1 which begins below-right). MSE scores are reported as 100 times their raw values to ease comparison with the line drawing case.

**Online visual discrimination.** This case differs from the others as it is a classification (or decision-making) task, where at the end of the presentation of the MNIST image the model must decide which number the digit belongs to (between 0 and 9). Since the decision is only made at the end of the sequence and targets are unavailable at intermediate points, this is a task with hard temporal credit assignment. The output of the model is a vector with probabilities of size 10 (one entry for each number), and the model was trained to maximise the likelihood of the target number using a standard cross-entropy error function.

**Visual-language task.** The architecture for the caption generation task consists of a pretrained convolutional neural network (CNN) coupled with an RNN (LSTM). The cerebellar network only communicates with the LSTM. The LSTM network has one layer of 256 LSTM units and the cerebellar network has two hidden layers (i.e. here we explicitly model a layer of Granule Cells and one of Purkinje Cells) of 1024 neurons.

The process from image to model-generated caption follows previous work[93] and is described next. As part of image preprocessing and data augmentation, which helps prevent model overfitting, a given image is randomly cropped to size $224 \times 224$, flipped horizontally with even chance, and appropriately normalised to be given to a pretrained Resnet model[94]. A feature vector $\mathbf{X}$ of size 256 is then obtained and passed to the LSTM at timestep 0. The LSTM is subsequently presented the "gold standard" caption $\{w_i\}_{i=1}^n$ one word per timestep, each time learning to predict the next word; i.e., at the timestep $t$ the model learns $P(w_t|\mathbf{X}, \{w_i\}_{i=1}^{t-1})$. The network simultaneously learns a word embedding so that each word $w_i$ is first transformed to a feature vector of size 256 before being given as input to the LSTM (as illustrated in (Fig. 8a). With a preset vocabulary of 9956 distinct words, the final output of the model ($P(w_i)$) is a probability vector of size 9956.

We found the models to be generally prone to overfitting the training data. For this reason, we apply dropout (during training Srivastava et al.[95]) on the input to the LSTM, where a given input element is set to zero with $p = 0.5$ probability. Once training is complete the models can generate their own captions to previously unseen images (Fig. 8, S17). Given an image at timestep 0, the model output at timestep $i$ is the word with the highest probability, and the same word is then provided as input to the model at timestep $i + 1$. In this way, the model can autonomously output an entire sequence of words which forms a predicted caption. In the (highly) rare case where the model generates a sequence of > 20 words, we consider only the first 20 words as its caption.

We used the COCO training data set ILSVRC-2012-CLS. This is a commonly used dataset available for our purposes under a Creative Commons license[55], which holds 414,113 image-caption pairs with 82,783 unique images while the held-out validation set (used for Fig. 8b, c) holds 202,654 with 40,504 unique images; note that each image, therefore, has ~5 distinct gold standard captions. Training takes place in batches of 100 image-caption pairs, with a learning rate of 0.001. Model performance is averaged over 10 random seeds. The performance is quantified in bits per word, which measures how good the model is at predicting the validation set. More specifically if a model assigns high probability to the test set (low BPW) it means it is not surprised by it hence indicating a good understanding of the language.

In order to judge the models beyond their learning curves in BPW, we quantify their ability to generate captions using a variety of language modelling metrics popular in the field of language evaluation. In particular, we compare model-generated captions against the gold standard captions using standard metrics in language modelling. We use the Semantic Propositional Image Caption Evaluation or SPICE metric referred to as caption score. This metric has been shown to be more accurate as it better captures the semantic structure of the generated captions[57]. We compare the SPICE metric across different groups of model-generated caption lengths (Fig. 8e), which we categorise as *short* (9 timesteps or less, ≤1.8 s), *medium* (between 10 and 13 timesteps inclusively, 2.3 s), and *long* (14 timesteps or more, >2.8 s). For both cRNN and cRNN, these caption lengths roughly comprise 39%, 59% and 2% of the total generated captions respectively.

Our code implementation is based on https://github.com/yunjey/pytorch-tutorial/tree/master/tutorials/03-advanced/image_captioning. To avoid issues with copyrights the photos illustrated in Figs. 8 and S17 were replaced with artificial ones generated using copyright-free DeepAI.org.

## Point-mass model
In order to demonstrate the capacity of ccRNN in a more realistic motor output setting, we evaluate the model's ability to perform the simple line drawing task using a simple physics point-mass motor model. In this setting, the model must produce a movement with a non-zero mass $m$. One could consider it to represent an hand or the pen/cursor that it manipulates[96].

Instead of the model directly predicting 2D coordinates as in the other drawing tasks, this model predicts a pair of orthogonal forces ($F^x$, $F^y$) which drive the point mass object along the $x$ and $y$ axis, respectively. The object then obeys the classical laws of motion. We describe the motion dynamics along the $x$ direction (the $y$ direction is analogous). Given the initial coordinate $x_{t-1}$ and the velocity $v_{t-1}^x$ of the object in the $x$ direction, the application of model-propelled force $F_t^x$ results in the following dynamics:

$$a_t^x = \frac{F_t^x}{m}$$
$$s_t^x = v_{t-1}^x \cdot \Delta t + \frac{1}{2} \cdot a_t^x \cdot (\Delta t)^2$$
$$x_t = x_{t-1} + s_t^x$$
$$v_t^x = v_{t-1}^x + a_t^x \cdot \Delta t$$

where $a_t^x$, $s_t^x$ represent the acceleration and displacement of the point mass object at time $t$, respectively. The motion dynamics are discretised into time windows of length $\Delta t = 0.1$ s. The initial coordinate $x_0$ and velocity $v_0^x$ are both set to zero and a mass $m = 0.1$ kg is used.

We apply these dynamics to the same task setup as in the simple line-drawing task. That is, the model must learn to translate external input at the first timestep to an associated temporal trajectory of the point mass object. To predict the forces ($F_t^x$, $F_t^y$) the RNN also receives as input the prior coordinates ($x_{t-1}, y_{t-1}$) and speed ($v_{t-1}^x, v_{t-1}^y$) of the object. As in the simple line drawing task, the task error at the current timestep is $E = ||(x_t, y_t) - (\hat{x}_t, \hat{y}_t)||_2$ where ($\hat{x}_t, \hat{y}_t$) is the target coordinate. To obtain the cerebral feedback signal (i.e. gradient) of the task error with respect to the model-applied force, i.e. $\frac{\partial E_t}{\partial F_t^x}$, we backpropagate through the motion dynamics above. We limit this backpropagation through the dynamics only to the immediate impact of the model force on the coordinate of the current timestep. The learning rate for this task is set as 0.003.

## Demixed principal component analysis
To study the response dynamics specific to task variables we perform demixed principal component analysis (dPCA)[97]. Demixed PCA extracts low-dimensional components that explain maximum population variance constrained by task-specific variables, such as the input stimulus. As a result we obtain principal components that are specific to task variables. The simulated neural data we provide as input to dPCA is a three-dimensional array ($n, s, t$) with neuronal activity across seeds, stimulus identity and time, respectively. In order to compare the cue-specific variance explained for each principal component across models we normalise against the variance explained for each principal component.

## Linear regression analysis

Linear regression was performed to determine how much variation of the unit activities in the hidden layer (i.e. Granule cells) of the cerebellar component is explained by the inputs, targets (i.e. external feedback) or true cerebral feedback signals. In particular, the activities of the hidden cerebellar component $\mathbf{Y}$ are modelled as

$$\hat{\mathbf{Y}} = \boldsymbol{\beta}_0 \mathbf{X} + \boldsymbol{\beta}_1 \qquad (16)$$

Where $\hat{\mathbf{Y}}$ are the predicted activities, $\mathbf{X}$ is the feature matrix which is one of the inputs, targets or true cerebral feedback signals, $\boldsymbol{\beta}_0$ is the fitted coefficient of these features, and $\boldsymbol{\beta}_1$ is a fixed bias term. This function is fitted using an ordinary least-squares method to minimise the residual sum-of-squares between the predicted activities $\hat{\mathbf{Y}}$ and observed activities $\mathbf{Y}$. These activities are sampled across the time span of each task at the start of each cerebral horizon window. For the simple line-drawing visuomotor task the cue (the only non-zero input) provided at the first timestep is used as the input to regress with throughout the task sequence. We fit this linear model at each epoch during training for each of the 10 random initialisations. Results are presented as averages with error bars.

## Measuring cerebro-cerebellar feedback similarity

The learning curves of ccRNN plotted against cRNN with a limited feedback horizon highlight the benefit of the feedback predicted by the cerebellar network. This indicates that the predicted feedback can indeed approximate the desired cerebral feedback. To verify this, we quantified the cerebro-cerebellar feedback similarity using cosine similarity − "cossimilarity" − between the predicted feedback and the optimal temporal cerebral feedback (as derived by gradient descent). Specifically given two arbitrary vectors $\mathbf{x}$ and $\mathbf{y}$

$$\text{cossimilarity}(\mathbf{x}, \mathbf{y}) = \frac{\mathbf{x} \cdot \mathbf{y}}{||\mathbf{x}||_2 ||\mathbf{y}||_2} \qquad (17)$$

where $\mathbf{x}$ is the predicted feedback and $\mathbf{y}$ the true optimal feedback, $\cdot$ denotes the dot product, and $||||_2$ is the Euclidean norm.

It is important to emphasise that true feedback is never actually provided to the model (as it goes beyond the feedback horizon $K$ considered). Instead, the cerebellum only learns through a combination of cerebral feedback within horizon $K$ and a bootstrapped term (see details above). This measure allows us to evaluate how much information about this ideal feedback can the cerebellum approximate. The final result is shown in Fig. 5a. To provide the reader with intuition about how feedback information degrades we highlight cases of external feedback available just at the end, which would lead to a gradual loss of the ability of the cerebellum to make good predictions for earlier points in the task. In particular, we highlight two task variants in which the task error is only defined at the end: visual discrimination and a simple line drawing variant where the external task feedback is only provided at the end of the task.

## Statistical analysis

Because the initial conditions of these types of models influence their learning trajectory we run our models across 10 different randomly chosen seeds. For all relevant figures except in Figs. S5–S7, significance was tested using a two-sided paired $t$-test across the different seeds on the relative changes; significance levels are represented as *($p < 0.05$), **($p < 0.01$), ***($p < 0.001$) and **** ($p < 0.0001$). For Figs. S5–S7, we apply a one-sided $t$-test as in ref. 97 (see figure legends for details); * denotes $p < 0.001$.

## Reporting summary

Further information on research design is available in the Nature Portfolio Reporting Summary linked to this article.

## Data availability

We have used standard machine learning data sets: the MNIST Dataset of handwritten digits (http://yann.lecun.com/exdb/mnist/) and COCO ILSVRC-2012-CLS training data set (https://cocodataset.org). Note that the example images shown in Figs. 8 and S17 were generated with deepAI.org to illustrate the visual context and are copyrights free. Source data are provided with this paper.

## Code availability

We used the PyTorch library for all neural network models. Our deep learning implementation is based on that of github.com/koz4k/dni-pytorch. The code and respective simulated data used for our experiments are available at https://github.com/neuralml/ccDNI[98]. For demixed PCA we used the following implementation https://github.com/machenslab/dPCA.

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

## Acknowledgements

We would like to thank the Neural & Machine Learning group, Paul Anastasiades, Paul Dodson, Conor Houghton, Laurence Aitchison, Cian O'Donnell, James M. Shine, Max Jaderberg, Nadia Cerminara and Jasmine Pickford for useful feedback. We would also like to thank Samia Mohinta and Milton Llera Montero for their help with setting up model analysis and training. J.P. was funded by an EPSRC Doctoral Training Partnership award (EP/R513179/1), E.B. by the Wellcome Trust (220101/Z/20/Z), P.C. by the Wellcome Trust (209453/Z/17/Z) and R.P.C. by the Medical Research Council (MR/X006107/1). This work made use of the HPC system Blue Pebble at the University of Bristol, UK.

## Author contributions

E.B., J.P. and R.P.C. developed the computational model, E.B. and J.P. performed simulations, R.P.C. supervised the project and E.B., J.P., P.C., R.A. and R.P.C. drafted and revised the article.

## Competing interests

The authors declare no competing interests.
