## [Peer Review File · Nature Communications]

Cerebro-cerebellar networks facilitate learning through feedback decouplingREVIEWER COMMENTS

Reviewer #1 (Remarks to the Author):

This manuscript introduces a model of cerebro-cerebellar networks, whose critical component is a feedback decoupling element. The model improves learning e.g. in a simple sensorimotor task in a cerebro-cerebellar recurrent neural network (ccRNN) configuration as opposed to a mere cerebral RNN (cRNN). Quite honestly, I am a bit puzzled by this work. There are three main limitations that truly limit the value of the model in its current description:

- 1) Figure 2, panel b (and others): inclusion of the cerebellar loop in the network certainly reduces performance errors (ccRNN vs cRNA). However, this 'smoothing' of performance in a sensorimotor task is found in literally every cerebellar model and is standard textbook matter for the last 50 years or so. In other words, the authors do not make it clear, what specifics of their model provide advantages and how their model compares to more classic models of cerebellar function / contribution. This could be done by not only showing cRNN and ccRNN results, but also run a more traditional model and show those results. Based on the claims in this manuscript, it is expected that the ccRNA model outperforms others.
- 2) It is not made clear how the ccRNN network model integrates true sensory feedback as opposed to predicted feedback. Experimental work in awake, behaving animals has shown that all cellular elements of the cerebellar system respond to all sensory modalities. This needs to be considered in an appropriate way.
- 3) Figure 4: the model cannot be tested against experimental data if parameters are not provided in absolute terms. In this figure, for example, the task feedback interval is given as a percent value. Is it not possible to provide an absolute time measure?

Reviewer #2 (Remarks to the Author):

In the present paper, Boven et al. suggest the idea that the main function of the cerebellar circuit is to provide a synthetic or simulated gradient (ie. feedback signal) to the neocortex, which promotes learning if feedback is sparse or only available at the end of the movement. Through a series of simulation studies, the authors explore some tasks where these advantages are present and present some findings of the learning dynamics in cortex and cerebellum.

Overall, I found the basic idea presented in the paper highly interesting, and think that this could be an very important paper for the cerebellar research community. On the other hand I found that the simulations and exposition had substantial weaknesses that currently limit the usefulness of the paper.

Main comments:

1. The main weakness of the paper is the relatively poor description of the tasks that underlie the simulations. Many details are not clear (see minor comments), and the specific tasks are very poorly motivated.
2. The reaching (or line drawing tasks) are simulated in 10 time steps and with a position-dependent output, which seems to have to match a motion of constant velocity. This would be ok as a toy model, but I believe that if the authors wish to claim that this circuit can give rise to “ataxia-like” symptoms, a little more effort for not being completely divorced from biology is required. At the very least, the authors should have the network provide muscle-like output (with an x- and y- actuator) and move a point mass with inertia. Of course there are also biologically arm models, but at least moving to this minimal example would instill some confidence that this idea could indeed explain ataxia.
3. The main variable variables of interest – i.e. the time points of feedback and the cortical feedback horizon need to be much better explained and more systematically studied, even using the 10-step toy model that you begin with. These concepts are the fundamental insight of the paper, but currently are obscured by a rapid succession of simulation and poor description of the tasks.
4. The online tasks , in which MNIST letters are sequentially presented have me puzzled. When combined with a reaching task, is this just a combination of the simple reaching task with the online visual discrimination? If yes, why? The main motivation is to present a more “complex and realistic” example? Complexity for the sake of complexity is worth very little – and it is not clear why these tasks are more realistic.

Overall I would recommend rethinking carefully which task simulations to present (and in which sequence). The most informative simulations are the ones where you show under which circumstances the ccRNN provides an advantage over the cRNN and where not (i.e. like in Figure 2d). For the MNIST task, I would expect that a ccRNN would not provide any advantage if the whole MNIST letter was presented at the same time – and the task made difficult by randomly degrading training stimuli? This would be an informative simulation that would actually make testable predictions. In contrast, I am not sure what I am supposed to learn from the online LD and DD tasks.

5. The choice of the caption-writing task is again very poorly motivated. The task is not immediately similar to tasks used to show language deficits in cerebellar patients. I do believe that the example and the analysis in Figure 8 d,e is interesting, but it is currently very hard to understand. The description of the learning structure should be improved and concentrate on the essential elements that tell us something about the boundary conditions under which the synthetic gradients are needed.

Minor comments

Line 58: I am not sure how the M/N ratio of 4 is justified. It is true that each GC receives 4-5 mossy fiber inputs. But there are ~40 million axons that exit the neocortex through the cerebral peduncles (Tomasch, 1969), which project on ~50 billion granule cells (Azevedo et al., 2009). Thus, the information expansion is likely more on the order of 1:1,000. It is true that most cortical neurons do not project to the cerebellum – but the assumptions should be clarified here.

Methods: line 512-540: The exposition of the general idea of synthetic gradients in the context of BPTT should be improved. For a Neuroscience audience, it may be good to first spell out equation 1 for the standard BPTT, and expose the problems of locking (i.e. wait for the update until time point T). Because it is important later, I would also first introduce a limited horizon BPTT (and motivate why this maybe a constraint for the nervous system) before introducing synthetic gradient idea. In equation 1, please clarify what time points in the process t, K, and T refer to. I assume K is the moment of update....

Example 1: A number of important details should be mentioned in the results. What is the input and when is it delivered? What is the output of the network (I assume it is x and y position?)? What is the desired trajectory (in time)? A linear motion at constant velocity? A minimal jerk trajectory? How many feedback points along the trajectory are given? How is the feedback delayed? These details are mostly in the methods, but are quite important to understand the simulations.

Line 100ff: The cerebral feedback horizon appears to be critical to understand the function of the cerebellum. However, a better explanation of the updating schedule and the points at which gradients are computed and back propagated is essential. I assume without a cerebellum, the gradient is computed and backpropagated every time a feedback point occurs? The truncation in Equation 1 is going forward in time (i.e. future error signals are being truncated), but conceptually you can also think about truncation of backpropagation running backwards in time. The relationship between the two should be explained more clearly. Also, does 50% mean 50% of the entire movement length (T)? How long are the gaps (and delay) of feedback points in this example?

Figure 2: What are the dashed lines in Figure 2e (right panel)? I assume that it constitutes the neural trajectory in the first 2 dPCA components over time? Is this based on cortical units only? How and when is the imperative stimulus provided to the network?

Figure 2 caption: “Euclidean distance between the two leading cue principal components”. This does not make sense to me (a Euclidian distance between 2 orthonormal vectors would always be same). Do you mean to say “Euclidean distance between neural state associated with the 7 target after projected onto

the first 2 dPCA components (as the main text seem to indicate)? If yes, why do the projection at all? Why not calculate the average Euclidian distance between the 7 targets across time using all data?

Figure 2f: Should the explained variance by the cue-related components not be tightly related to the distance between the different trial types (shown in Figure 2e)? What am I missing? Likely it is confusing as you split this up by different dimensions? Do the lineplots indicate the cumulative variance? If yes, this is not mentioned in the caption? What does the fact that the cRNN has more cue-related variance in the first two dPCA dimensions tell us?

Figure 2g: right panel: It is not clear what the different circles of the same color refer to. Why are there no dashed lines, as in Figure 2e?

Figure 2g: I assume that the separation between targets (or dimensions – again this is confusing) goes to zero, as the gradients vanish in the end of learning? Would the gradients persist if there was noise in the network that would need to be corrected?

Line 135: “in which sensory input was only provided at the start of the task” – this should be mentioned earlier (see above). Also, it seems you are referring to the imperative stimulus (x), rather than the task feedback ($E(t)$) here, correct?

Line 137: This make it sounds like that the online LD visuomotor task and the online DD visuomotor task also receive the raster-wise scan of the MNIST digits. This does not make much sense, unless you explain how the digits are related to the desired output.

Line 142: Unclear what is learned with the online LD visuomotor task over and above the last simulation.

Line 146: The motivation for the online visual discrimination task is not clear.

Line 623: The description of the simple visuomotor task is not clear. I assume x are the discrete cues – are these labeled lines or a single input unit with $\{-1,2,3\}$ activity level setting? How are the lines (y) defined (which angles?). Is the network doing the task in 10 time steps? At which time steps is feedback provided, and how?

Line 630: “the model receives the same temporal input”. As what? Not the simple VM task, as here you use MNIST digits?

Line 647: If the MNIST digits are associated with different spatial targets, does the network first have 28 time steps to identify the digit and then 10 time steps to produce the output.

Line 185: “For the simple LD task we observe that the predictions made during earlier points in the task are more similar than those at later points (Fig.5c). These results suggest....”.

I guess this behavior can *kind of* be seen in Figure 5d in the left panel, but not in any of the other panels. Should the effect (late time points represented early in learning, early time points late in learning) not be more pronounced in the end-only feedback tasks? The simulations presented here seem inconclusive in this regard.

Line 193: These are interesting predictions, but in the current form unfortunately vacuous. What is required is a much more careful analysis of which behavior to expect under which conditions.

Line 204: It seems that you somehow did PCA on the Time x Cortical units x Cerebellar Units tensor? Or one the Time x (cortical units * cerebellar units) correlation matrix? Please state explicitly here in the results.

Line 239: This is an interesting observation, but it would be nice to spell this out more closely. I assume that this is due to the fact that an inappropriate learning gradient now persists and cortical learning is thrown off its correct path?

Line 255: It is unclear from this what the teaching signal is – Are the words of the desired caption not a supervised teaching signal?

Cerebro-cerebellar networks facilitate learning through feedback decoupling

Response to reviewers

Dear Reviewers,

We would like to thank the reviewers for the positive and constructive feedback. We have addressed all raised points in detail. In the revised manuscript we have:

1. Included new simulations (new comparison with standard cerebellar models, new point-mass model, new exhaustive analysis of parameters of interest, new linear regression decoding analysis, new noise analysis);
2. Updated 5 main figures (Figs. 1, 2, 3, 4 and 8), including a new schematic to explain the differences between the multiple tasks;
3. Added new 7 supplementary figures (Figs. S2, S4, S8, S9, S10, S11, S13) including new results with new control models, and new conditions not tested before, new schematics, and others).
4. Rewritten substantial parts of the manuscript to clarify the points raised by the reviewers.

Overall, we believe that our paper has been greatly improved in the process.

Best regards,
Ellen Boven
Joe Pemberton
Paul Chadderton, PhD
Richard Apps, PhD
Rui Ponte Costa, PhD

Reviewers' comments:

Reviewer #1:

This manuscript introduces a model of cerebro-cerebellar networks, whose critical component is a feedback decoupling element. The model improves learning e.g. in a simple sensorimotor task in a cerebro-cerebellar recurrent neural network (ccRNN) configuration as opposed to a mere cerebral RNN (cRNN). Quite honestly, I am a bit puzzled by this work. There are three main limitations that truly limit the value of the model in its current description:

1) Figure 2, panel b (and others): inclusion of the cerebellar loop in the network certainly reduces performance errors (ccRNN vs cRNA). However, this 'smoothing' of performance in a sensorimotor task is found in literally every cerebellar model and is standard textbook matter for the last 50 years or so. In other words, the authors do not make it clear, what specifics of their model provide advantages and how their model compares to more classic models of cerebellar function / contribution. This could be done by not only showing cRNN and ccRNN results, but also run a more traditional model and show those results. Based on the claims in this manuscript, it is expected that the ccRNN model outperforms others.

Reply: Our model focuses on modelling cerebro-cerebellar interactions. To the best of our knowledge this is the first model to do so. We agree that our model could be better contrasted with classical models. To address this we have performed additional control simulations and rewrote parts of the main text to highlight the key differences between ccRNN and classical cerebellar models. Below we highlight three key points in which our model deviates and provides advantages from classical models:

1. Although classical Albus-Marr models of the cerebellum show a speed-up in the learning curves, they do not deal with temporal problems. Given that the problems faced by both animals and humans alike are all virtually of a temporal nature, we think that it is critical to have models that can learn temporal tasks. Our ccRNN has the capability of learning challenging temporal tasks (e.g. Fig. 3). To better demonstrate the benefits of our model we have performed a new set of simulations in which we show that classical Albus-Marr type of models, consisting solely of a feedforward cerebellar network, completely fail to learn temporal tasks (purple line in Fig. S4). Since our model considers cerebro-cerebellar interactions we also add a classical cerebral model to complete the comparison to more standard models. Therefore, we consider a model in which the cerebrum (RNN) is not plastic which performs substantially worse than our model (green line in Fig. S4). We discuss these results in lines 103-106.
2. In order for the cerebellar component of our model to facilitate learning in temporal tasks, it must learn efficiently. To achieve this the cerebellum learns using the cerebral feedback, but it also learns through self-predictions. This is a principle commonly used in temporal reinforcement learning, and that we propose to be also of importance for cerebellar learning. Without this component, for the cerebellum to learn it would have to wait until the true feedback is received from the cerebrum,

which would defeat the point of our cerebellar model, i.e. having effective temporal feedback predictions. We highlight this component in Fig. 1 (panel b), and it is also explained in lines 75-77, 168-169 and discussed in lines 313-317. Moreover, we have now conducted an additional set of simulations without self-prediction learning to show that this is an important component in the model (Fig. S13).

3. Finally, sparse task feedback is an inherent property of virtually any task. In our model this sparse feedback results in dysmetria-like behaviours in models without a cerebellum. To the best of knowledge this provides a new possible explanation for how dysmetria can develop through cerebrum-cerebellar interactions, not just in the motor domain (e.g. Fig. 2-3) but also in the cognitive domain (Fig. 3,8). Moreover, in the tasks we consider classical Albus-Marr type of models show worse dysmetria than the cerebral network (Fig. S4). This goes back to the point we made above, in which we highlight that these classical models cannot deal (per se) with temporal tasks.

2) It is not made clear how the ccRNN network model integrates true sensory feedback as opposed to predicted feedback. Experimental work in awake, behaving animals has shown that all cellular elements of the cerebellar system respond to all sensory modalities. This needs to be considered in an appropriate way.

Reply: Our model integrates both predicted and true sensory feedback. To help clarify what is considered as true and predicted sensory feedback we have included these terms in Fig. 1:

First, the cerebral RNN (i.e. a cerebral area) relies on **true sensory feedback** (red top right arrow) to generate its own **feedback error signals** (i.e. **cerebral feedback**). The classical example in the brain of such prediction errors are those computed by the VTA (reward-prediction errors) (Schultz et al. Science 1997). More recently, feedback information akin to the ones used by the

model have been thought to be present throughout the brain (Banerjee et al. Nature 2020). Second, the cerebellum uses **cerebral feedback** (red arrow going to inferior olive) to learn to **predict cerebral feedback** (blue arrow going out of the cerebellum), thereby speeding up learning and reducing dysmetria like behaviours.

Regarding your second point, to show that the cerebellum responds to all sensory modalities we perform a task-encoding analysis of cerebellar neural activity (Fig. 2h-f). This shows that the cerebellum does encode information about all sensory cues (note that these cues can be from any sensory modality), each being directly related to both sensory input itself, but also true sensory feedback, as each cue has an associated target trajectory. Moreover, in addition to the demixed PCA analysis, we now include a linear regression analysis to provide

further evidence of cerebellar activity containing information about *sensory inputs* (i.e. cues), *sensory feedback* (i.e. targets) and internally generated *cerebral feedback* (line 133; 823-835). Our new analysis shows that the model contains information about all three in both the simple line drawing task (Fig. S8a) and the online line drawing task (Fig. S8b). Note that the amount of information depends on the learning stage (compare with Fig. 2). Consequently for more naturalistic tasks, as the ones shown in Fig. 3, that are never fully learnt this means that the cerebellum contains information about sensory input and feedback even after learning has converged. Importantly, as these tasks can represent any modality (e.g. auditory cue vs visual cue), these results are consistent with experimental observations, as highlighted by the reviewer.

3) Figure 4: the model cannot be tested against experimental data if parameters are not provided in absolute terms. In this figure, for example, the task feedback internal is given as a percent value. Is it not possible to provide an absolute time measure?

Reply: Thank you for pointing this out. This is indeed something that we had considered before. We have now added a new supplementary section which explains how time can be explicitly considered in the model (lines 626-635). In line with others (Song et al., 2017; Yang et al, 2019), we assume each timestep to correspond to 0.1s (and 0.2s for the language task). As a result we have updated all temporally relevant x-axis across tasks in all the figures in the paper.

Reviewer #2:

In the present paper, Boven et al. suggest the idea that the main function of the cerebellar circuit is to provide a synthetic or simulated gradient (ie. feedback signal) to the neocortex, which promotes learning if feedback is sparse or only available at the end of the movement. Through a series of simulation studies, the authors explore some tasks where these advantages are present and present some findings of the learning dynamics in cortex and cerebellum.

Overall, I found the basic idea presented in the paper highly interesting, and think that this could be an very important paper for the cerebellar research community. On the other hand I found that the simulations and exposition had substantial weaknesses that currently limit the usefulness of the paper.

Main comments:

1) The main weakness of the paper is the relatively poor description of the tasks that underlie the simulations. Many details are not clear (see minor comments), and the specific tasks are very poorly motivated.

Reply: Thank you for raising this lack of detail. Although we provide the details in the supplementary material we agree that we were somewhat short in the description and motivation of the tasks in the main text. We have now added further details in the main text for the different tasks, in particular to the more complex tasks (see also other points raised below). We have also added a new schematic that follows the same structure across all the tasks such that it is easy for the reader to understand the input-output transformations that the models must learn to perform for each task. Below we provide a summary of the tasks and their motivation:

1. Line drawing sensorimotor task: This task is inspired by classical sensorimotor studies in the cerebellum, which include line drawing and target reaching tasks (Butcher et al., 2017; Nashef et al., 2019; Sanes et al., 1990; Streng et al., 2018; Tseng et al., 2007). In this task the model must draw a straight line in a two-dimensional space towards one of seven target locations given a target-specific cue provided only at the start of the task (see schematic above and more details in lines 80-86 and new Fig. 2a).

2. Online line/digit drawing sensorimotor task: Most naturalistic sensorimotor transformations are more complex than drawing a straight line given a simple cue. Both animals and humans need to map rich nonlinear spatiotemporal sensory signals onto motor outputs (Pouget and Snyder 2000 Nature Neurosci., Dennis et al. 2021 JNeurosci.). For example, navigating a forest, riding a bike or playing tennis. The first step we took in this direction was to consider a line drawing task with non-linear continuous input (green line on the schematic), which is in contrast with the simple sensory input received only at the beginning in the simple line drawing task. However, the online LD task considers only nonlinear inputs, whereas the outputs are linear. For this reason, next we considered a task in which we train the network to draw a nonlinear trajectory (digit drawing) given a nonlinear input (this has been expanded upon in lines 140-146 and new Fig. 3a).

3. Online visual discrimination task: There is growing evidence suggesting that the cerebellum is also involved in non-motor tasks (Fiez et al 1992, Baker et al. 1996, Guell et al. 2015, Brissenden et al. 2019, Schmahmann et al. 2019, Guell et al. 2018, Devereitt et al. 2019). To test whether our observations in the sensorimotor tasks generalise to non-motor domains, while using the same input statistics as in the previous tasks we trained the model in a (online) visual discrimination task (this has been expanded upon in lines 158-160 and new Fig. 3a).

4. Visual-language modelling task: This task was chosen to illustrate that our model is applicable to a wider range of cognitive tasks while being inspired by a body of work showing language deficits in cerebellar patients (e.g. Stoodley and Schmahmann 2009 and Guell et al. 2015). More specifically, we model the *recreating sentence task* studied by Guell et al. 2015 in cerebellar patients, which showed a poor semantic description of images when compared to healthy subjects. To demonstrate a similar behaviour in our model we built on existing datasets commonly used in caption generation tasks in machine learning. In this task the model is provided with a low dimensional representation of a natural image and the RNNs are then trained to consequently predict the next word given the compressed sensory input and the previous word (see schematic above; this has been expanded upon in lines 268-271 and in the new Fig. 8a).

2) The reaching (or line drawing tasks) are simulated in 10 time steps and with a position-dependent output, which seems to have to match a motion of constant velocity. This would be ok as a toy model, but I believe that if the authors wish to claim that this circuit can give rise to “ataxia-like” symptoms, a little more effort for not being completely divorced from biology is required. At the very least, the authors should have the network provide muscle-like output (with an x- and y- actuator) and move a point mass with inertia. Of course

there are also biologically arm models, but at least moving to this minimal example would in still some confidence that this idea could indeed explain ataxia.

Reply: Thank you for the suggestion. We have now extended our model to include a point-mass model of the motor output (details of this model are described in lines 799-814). We have performed a new set of simulations of the line drawing task using this model (see new Fig. S2 and lines 85). As expected this new model makes learning more challenging to the model, but the relative comparison between cerebellar and non-cerebellar models remains the same in terms of (i) faster learning (Fig. S2a), (ii) reduced ataxia (Fig. S2a,b), (iii) dependency on feedback horizon (Fig. S2c) and (iv) dependency on task feedback sparsity (Fig. S2d,e). These results demonstrate that the key model predictions also apply to a more realistic motor model.

3) The main variables of interest – i.e. the time points of feedback and the cortical feedback horizon need to be much better explained and more systematically studied, even using the 10-step toy model that you begin with. These concepts are the fundamental insight of the paper, but currently are obscured by a rapid succession of simulation and poor description of the tasks.

Reply: We now provide a new schematic for all the tasks that should help to clarify their differences and similarities in terms of input-output transformations and feedback provided (see point above, and the respective main figures: Fig. 2a, 3a, 8a). In addition, we have now conducted a systematic study of the interaction between feedback interval (sparsity) and horizon, which is now included in the main figure (Fig. 4d). There are some minor differences in the new Fig. 4 compared to the previous one. This is simply due to a minor discrepancy that we had in terms of sampling external feedback which we have now corrected. This more systematic analysis is in line with the results we have already reported, in that the ccRNN model particularly improves learning and dyslexia-like outputs for medium-to-hard regimes of feedback interval, provided that the cerebral feedback horizon is not longer than the feedback interval. This is to be expected, as cerebellar feedback in our model would be mostly beneficial for points in time in which there is no feedback readily available (lines 182-186).

The reviewer points out that these are the “fundamental insights from this paper”. While we agree that these are fundamental insights, we also that there are a number of other observations/predictions that we think to be important. We highlight three other main ones below:

1. We make predictions in terms of how cerebral-cerebellar task-specific representations co-evolve in our model (Fig. 2e-h).
2. Our model requires the cerebellum to be trained with feedback from the cerebral cortex, but also with its own self-predictions (also known as bootstrapping). This idea is highlighted in Fig. 1b and is inspired by the reinforcement learning literature. In the new version of the manuscript we now include new simulations showing that this is critical for the cerebellum to learn to provide effective feedback (Fig. S13). This reinforces the idea that the cerebellum must learn fast in order to provide effective

feedback to the rest of the brain. We think this is a new insight that will also guide future theoretical and experimental work.

3. Using ablation studies we demonstrate that the cerebellum is important not only early in learning, but also after convergence in cases in which the model does not reach optimal performance (i.e. $\text{error} > 0$). Interestingly, our model predicts that if the inferior olive-like structure is ablated then this completely abolishes performance across all tasks even after learning has converged. This is because the cerebellum will corrupt the learning trajectory of the cerebral RNN. There is some very recent experimental evidence to suggest that this is the case (Silva et al. bioRxiv 2022; we now highlight this in line 263).

4) The online tasks, in which MNIST letters are sequentially presented have me puzzled. When combined with a reaching task, is this just a combination of the simple reaching task with the online visual discrimination? If yes, why? The main motivation is to present a more “complex and realistic” example? Complexity for the sake of complexity is worth very little – and it is not clear why these tasks are more realistic.

Overall I would recommend rethinking carefully which task simulations to present (and in which sequence). The most informative simulations are the ones where you show under which circumstances the ccRNN provides an advantage over the cRNN and where not (i.e. like in Figure 2d). For the MNIST task, I would expect that a ccRNN would not provide any advantage if the whole MNIST letter was presented at the same time – and the task made difficult by randomly degrading training stimuli? This would be an informative simulation that would actually make testable predictions. In contrast, I am not sure what I am supposed to learn from the online LD and DD tasks.

Reply: Thank you for these comments and suggestions. We will address each point in order:

1. Use of MNIST dataset: We hope that the details provided in the first major point and the respective changes (text and new schematics) help clarify the key differences between tasks. Using the MNIST dataset allows us to test the model with nonlinear input-output mappings. The visual discrimination task allows us to make a first link with cognitive tasks, but also tests a case in which the feedback is only provided at the end (Fig. 3a), rather than throughout the task, which is in contrast to all the tasks tested until that point, which use sparse feedback throughout the task. We have clarified this in line 158-160. We have also clarified the overall motivation for the different tasks (see Major Point 1).
2. Complexity: We would like to point out that the motivation for these more challenging tasks is not complexity per se. But rather trying to be closer to naturalistic conditions by using nonlinear input-output mappings. We now show a new schematic across figures for the multiple tasks that makes this point more clear. To help clarify the differences in input structure we also provide a new supp. figure with a few input samples for both the simple and online LD tasks (Fig. S9). These more challenging/realistic tasks also make the important point that when the model converges to a non-zero error solution (e.g. online LD and DD tasks) the cerebellum is still important, which is not the case in the simpler/toy LD task (see our ablation results that highlight this point Fig. 7a,b).

3. New simulation with different levels of noise: We agree it would also be interesting to test ccRNN with degrading stimuli. To this end we have now added a figure (line 152; Fig. S10) which shows how the model performs under various degrees of (Gaussian) noise in the input images; as expected ccRNN copes well under modest noise levels both models break down for high enough noise. Interestingly, we find that the cerebellum helps slightly more with non-zero noise.
4. New simulation with whole MNIST: We agree with the reviewer that by showing the full image at once, then there is no need for our cerebellar module as there is no need for temporal credit assignment. This is indeed what we find and have tested a range of input lengths to demonstrate this behaviour (lines 155-157; Fig. S11).

5) The choice of the caption-writing task is again very poorly motivated. The task is not immediately similar to tasks used to show language deficits in cerebellar patients. I do believe that the example and the analysis in Figure 8 d,e is interesting, but it is currently very hard to understand. The description of the learning structure should be improved and concentrate on the essential elements that tell us something about the boundary conditions under which the synthetic gradients are needed.

Reply: Thanks for pointing out this lack of clarity. We have chosen this task as a proof of concept that our model is readily applicable to what are usually referred to as high-level cognition. Moreover this builds on several studies showing language deficits in cerebellar patients (e.g. Stoodley and Schmahmann 2009, Guell et al. 2015 Guell et al. 2018 and Silveri 2021). In particular, we model the *recreating sentence task* studied by Guell et al. 2015, which showed a poor semantic description of images in cerebellar patients when compared to healthy subjects. An example (reproduced

from Fig. 3 in Guell et al. 2015) of an image for the supermarket topic used in this experiment is given above. A description written by a cerebellar patient for this image was: "The carrots were not available neither this week nor next", which is not semantically quite accurate (Table 3 Guell et al. 2015).

Overall cerebellar patients showed lower scores in this type of tasks (Guell et al. 2015). To model this task in our model we use existing caption-generation datasets from machine learning, which have a similar nature, in that an image is presented as input and the model is trained to generate a textual description of that image. Multiple examples of images-text pairs used to train our model are given in Fig. 8c and Fig. S18. As described in detail in Point 1 above in this task the model is provided with a low

dimensional representation of an image (e.g. bird, see schematic) and then the RNN is trained to predict the next word given the visual sensory input and the previous word (this has been expanded upon in lines 268-271 and in a modified Fig. 8a). In terms of the “boundary conditions” this refers to the feedback horizon that defines how far back in time the true cortical feedback (i.e. gradients) is available. Consequently the feedback estimated by the cerebellum would provide useful gradients from later target words (i.e. *large* onwards) back to the sensory input (for the case of feedback horizon = 1 word) as illustrated in the schematic above. This ability to use estimated feedback further back in time provides an explanation for the experimental observations of Guell et al. 2015: cerebellar patients have poorer temporal credit assignment, therefore are not able to capture the complex sensory-language dependencies that are required for these tasks as well as healthy subjects (we discuss this in lines 279-282).

Minor comments

1) Line 58: I am not sure how the M/N ratio of 4 is justified. It is true that each GC receives 4-5 mossy fiber inputs. But there are ~40 million axons that exit the neocortex through the cerebral peduncles (Tomasch, 1969), which project on ~50 billion granule cells (Azevedo et al., 2009). Thus, the information expansion is likely more on the order of 1:1,000. It is true that most cortical neurons do not project to the cerebellum – but the assumptions should be clarified here.

Reply: We should clarify that this ratio is simply about the total number of cells in the cerebrum (16 billion) versus the cerebellum (69 billion), not about the GC's highly sparse connectivity. We used the same cerebrum/cerebellar ratio estimated experimentally as reviewed by Herculano-Houzel Frontiers 2009. We detail this in lines 58-60.

2) Methods: line 512-540: The exposition of the general idea of synthetic gradients in the context of BPTT should be improved. For a Neuroscience audience, it may be good to first spell out equation 1 for the standard BPTT, and expose the problems of locking (i.e. wait for the update until time point T). Because it is important later, I would also first introduce a limited horizon BPTT (and motivate why this maybe a constraint for the nervous system) before introducing synthetic gradient idea. In equation 1, please clarify what time points in the process t , K , and T refer to. I assume K is the moment of update....

Reply: Thanks for this suggestion. We agree that this would indeed be useful. We have now revised this section to first introduce full BPTT, its limitations and then gradually introduce truncated BPTT before introducing the idea of synthetic gradients (see lines 579-603). t , K , T refer to the current timestep, the imposed horizon for BPTT, and the total duration of the task respectively (this is also clarified in lines 579-587). At K the gradient information is available so it is possible to update at this point. However, to ensure a fairer comparison between ccRNN and cRNN we have decided to accumulate all updates and perform the update at the end (lines 670-675).

3) Example 1: A number of important details should be mentioned in the results. What is the input and when is it delivered? What is the output of the network (I assume it is x and y position?)? What is the desired trajectory (in time)? A linear motion at constant velocity? A minimal jerk trajectory? How many feedback points along the trajectory are given? How is the feedback delayed? These details are mostly in the methods, but are quite important to understand the simulations.

Reply: Thanks for this suggestion. The new schematic should help clarify this (e.g. Fig. 2a) and we have also added a few clarifications in the main text (e.g. lines 80-81, 84-86). For the motor tasks we model directly x,y coordinates at each timestep (but see also the new model as described in major point 2 above). The desired trajectory is given in the figures (Fig. 2a,b and Fig. 3a as dashed grey lines) and the external feedback is represented as red dots along this line. The delay of the feedback with respect to the input is now made clearer with the new schematics (Fig. 2a and Fig. 3a).

4) Line 100ff: The cerebral feedback horizon appears to be critical to understand the function of the cerebellum. However, a better explanation of the updating schedule and the points at which gradients are computed and back propagated is essential. I assume without a cerebellum, the gradient is computed and backpropagated every time a feedback point occurs? The truncation in Equation 1 is going forward in time (i.e. future error signals are being truncated), but conceptually you can also think about truncation of backpropagation running backwards in time. The relationship between the two should be explained more clearly. Also, does 50% mean 50% of the entire movement length (T)? How long are the gaps (and delay) of feedback points in this example?

Reply: Thanks for raising this lack of clarity.

1. *Updating schedule*: As stated in a previous minor point we already described this in the experimental details (now expanded in lines 669-675), but we have now added a note on this after Equation 2 (see lines 601-603).
2. *Points in which gradients computed*: Yes, without a cerebellum BPTT gradients are computed every time a feedback point occurs (see new Equation 1). The summations (with index t') are going forward in time, but the gradients are always calculated backward in time, from a given point t' to a previous time point. We have now clarified this in lines 579-603.
3. *With respect to the percentages* used to describe the length e.g. cerebral feedback horizon (for example Fig. 3d) the reviewer is correct in that these are percentages of the entire movement length (T). To be able to relate better with experiments we now also provide these in actual time (lines 626-635)

5) Figure 2: What are the dashed lines in Figure 2e (right panel)? I assume that it constitutes the neural trajectory in the first 2 dPCA components over time? Is this based on cortical units only? How and when is the imperative stimulus provided to the network?

Reply: Yes, that is correct, and yes this is based on cortical units only (panels Fig. 2e-f apply only to cortical units). We now clarify this in Fig. 2 caption. As clarified below and above the

external sensory stimulus is only provided at the start, the new schematic in Fig. 2a clarifies this.

6) Figure 2 caption: “Euclidean distance between the two leading cue principal components”. This does not make sense to me (a Euclidian distance between 2 orthonormal vectors would always be same). Do you mean to say “Euclidean distance between neural state associated with the 7 target after projected onto the first 2 dPCA components (as the main text seem to indicate)? If yes, why do the projection at all? Why not calculate the average Euclidian distance between the 7 targets across time using all data?

Reply: Yes that is correct, we are computing the Euclidean distance between the seven cues of the RNN dynamics corresponding to the two leading dPCA components. We have now rephrased this (see new caption). We are working on this subspace to be able to relate the euclidean space directly to the dPCAs shown on panel 2e (right).

7) Figure 2f: Should the explained variance by the cue-related components not be tightly related to the distance between the different trial types (shown in Figure 2e)? What am I missing? Likely it is confusing as you split this up by different dimensions? Do the lineplots indicate the cumulative variance? If yes, this is not mentioned in the caption? What does the fact that the cRNN has more cue-related variance in the first two dPCA dimensions tell us?

Reply: Thanks for pointing this out. In order to make the explained variance of the cue latent information directly comparable between cRNN and ccRNN in Figure 2f, we are now calculating the normalised cue explained variance at the epoch where the Euclidean distance is maximal. This normalisation is needed because the variance explained by each component for the respective models is different. More specifically, we normalise the variance of each component in the cue marginalisation by the variance explained by each component for the respective model. We have now changed this in Figure 2f and Figure 2g and added a description in the respective captions as well as lines 126-127 in the main text and lines 823-825 in the Supplementary Materials and Methods. When now comparing this normalised cue-specific explained variance we can directly relate the distance to the explained variance by the cue-related component.

8) Figure 2g: right panel: It is not clear what the different circles of the same color refer to. Why are there no dashed lines, as in Figure 2e?

Reply: As the cerebellum is a simple feedforward network it does reflect temporal evolution of cue-specific information. The multiple circles reflect the different points in time throughout the task, and as can be seen there is no clear temporal trajectory. This is why we do not have the dashed lines. We have now clarified this in the caption.

9) Figure 2g: I assume that the separation between targets (or dimensions – again this is confusing) goes to zero, as the gradients vanish in the end of learning? Would the gradients persist if there was noise in the network that would need to be corrected?

Reply: Yes, the separation goes to zero because at the end there is no longer the need to encode gradients as the task has been fully learnt. This is part of the reason why we think it is important to have more complex tasks that make the point that for more realistic tasks the error is unlikely to reach zero, therefore the cerebellum would still encode task-relevant information even after learning has converged. The same would be true with noise.

10) Line 135: “in which sensory input was only provided at the start of the task” – this should be mentioned earlier (see above). Also, it seems you are referring to the imperative stimulus (x), rather than the task feedback ($E(t)$) here, correct?

Reply: Thanks for the suggestion. We now highlight this earlier on (see line 81). Yes, that is correct we always refer to the imperative stimulus as sensory input and the sensory feedback as feedback (or external feedback).

11) Line 137: This make it sounds like that the online LD visuomotor task and the online DD visuomotor task also receive the raster-wise scan of the MNIST digits. This does not make much sense, unless you explain how the digits are related to the desired output.

Reply: In both the online LD and DD visuomotor tasks the MNIST digits are also used as input (see details now provided in the major points above). The new task schematics should help clarify how the input-output are related for these tasks (Fig. 3a).

12) Line 142: Unclear what is learned with the online LD visuomotor task over and above the last simulation.

Reply: The principal motivation for the online LD visuomotor task is to directly observe how ccRNN can generalise to a more naturalistic task setting (with ongoing input, as discussed in major points 1,4 above). Importantly, with this more challenging task the model does never reach perfect performance (Fig. 3b), which means that the cerebellum is important even after learning convergence (see ablation results in Fig. 7), but see also more details on this in the major points above.

13) Line 146: The motivation for the online visual discrimination task is not clear.

Reply: We have now expanded our motivation (lines 158-160).

14) Line 623: The description of the simple visuomotor task is not clear. I assume x are the discrete cues – are these labeled lines or a single input unit with $\{+1,2,3\}$ activity level setting? How are the lines (y) defined (which angles?). Is the network doing the task in 10 time steps? At which time steps is feedback provided, and how?

Reply: The new schematic should help clarify this (see Fig. 2a and the multitask schematic above). To answer your questions specifically (see also lines 80-81, 84-86, 721-722):

1. *Are these labelled lines or a single input unit with $\{+1,2,3\}$ activity level setting?* The discrete cues correspond to a single input with $\{+1, 2, 3\}$ activity level setting.

Alternatively this can be viewed as a vector with the discrete cue occurring at the first time step and zero input at the remaining 9 time steps.

2. *How are the lines (y) defined (which angles?)* The lines are defined by creating 6 linearly spaced vectors between the origin and 6 end points that lie equidistantly on a circle centred on the origin with radius 10. The 7th target is defined as a vector of 0 for x and y coordinates, as to consider the case in which the network remains quiet at the centre of the drawing screen.
3. *Is the network doing the task in 10 time steps?* Yes, the network is performing the task over 10 time steps.
4. *At which time steps is feedback provided, and how?* External sensory feedback is provided every other time step by presenting the corresponding x, y coordinate.

15) Line 630: “the model receives the same temporal input”. As what? Not the simple VM task, as here you use MNIST digits?

Reply: The same temporal input is used in each of the online visuomotor tasks; that is, a row by row presentation of a given MNIST digit. This is not the same as the simple VM task which uses a cue provided at the start of the task, but we agree this sentence could be misleading. We have clarified this accordingly (lines 732-735).

16) Line 647: If the MNIST digits are associated with different spatial targets, does the network first have 28 time steps to identify the digit and then 10 time steps to produce the output.

Reply: Although it is in principle possible to produce the output only after the entire presentation of the image, and indeed what happens in the discrimination task, this is not the case in the online visuomotor drawing tasks the model learns as they produce the output as a function of the input received. That is, it has the same window as the input presentation - 28 timesteps - to produce the output. The new task schematics now clarify this (Fig. 3c).

17) Line 185: “For the simple LD task we observe that the predictions made during earlier points in the task are more similar than those at later points (Fig.5c). These results suggest....”.

I guess this behavior can *kind of* be seen in Figure 5d in the left panel, but not in any of the other panels. Should the effect (late time points represented early in learning, early time points late in learning) not be more pronounced in the end-only feedback tasks? The simulations presented here seem inconclusive in this regard.

Reply: In general we expect later feedback to be more important as that is also the feedback that should show stronger errors due to the fact that errors will accumulate throughout the task. This is clear in the end-only tasks (Fig. 5a,b) but also in both the online LD and online DD. In these end-only tasks we do not see “early time points with high similarity late in learning” because there is no sensory feedback at early points in the task (see lack of red arrows at the top of the plots) but also because of the cerebral feedback horizon being relatively short. This means that at those points the cerebellum can only rely on learning via self-prediction (i.e. bootstrapping) which is prone to noise/errors, thereby reducing the

similarity with the optimal cerebral feedback (as derived from full BPTT). We have rephrased this sentence (lines 201-208).

18) Line 193: These are interesting predictions, but in the current form unfortunately vacuous. What is required is a much more careful analysis of which behavior to expect under which conditions.

Reply: We have now made more specific predictions based on our results (see lines 210-213). In particular, (i) for tasks with feedback only at the end (Fig. 5a), the model predicts that cerebro-cerebellar feedback alignment should decay rapidly and (ii) for tasks with regular external feedback (Fig. 5c) the model predicts that cerebro-cerebellar feedback alignment should be stronger when more external feedback is provided.

19) Line 204: It seems that you somehow did PCA on the Time x Cortical units x Cerebellar Units tensor? Or one the Time x (cortical units * cerebellar units) correlation matrix? Please state explicitly here in the results.

Reply: The PCA analysis is done on the time x cerebro-cerebellar pairwise correlation coefficient matrix, as mentioned in caption of Figure 6b and rewritten in line 223. This is now added to section cerebro-cerebellar coupling in material and methods (see line 704 in Supplementary material).

20) Line 239: This is an interesting observation, but it would be nice to spell this out more closely. I assume that this is due to the fact that an inappropriate learning gradient now persists and cortical learning is thrown of its correct path?

Reply: The reviewer is correct, and in line with very recent observations (Silva et al. 2022 bioRxiv). We now state this on lines 259-264.

21) Line 255: It is unclear from this what the teaching signal is – Are the words of the desired caption not a supervised teaching signal?

Reply: In general language models are trained to predict future words, for which it is only required text that is per se not supervised (i.e. its only data), this is referred to as self-supervised (unsupervised) learning, in the field of machine learning. However, we agree that the captions used here were created by humans as labels or targets for these images, which makes it supervised. We have now removed this sentence.

REVIEWERS' COMMENTS

Reviewer #1 (Remarks to the Author):

My previous concerns have been appropriately addressed.

Reviewer #2 (Remarks to the Author):

In the present paper, Boven et al. suggest the idea that the main function of the cerebellar circuit is to provide a synthetic or simulated gradient (ie. feedback signal) to the neocortex, which promotes learning if feedback is sparse or only available at the end of the movement. Through a series of simulation studies, the authors explore tasks where these advantages are present and show findings of the learning dynamics in cortex and cerebellum.

The paper presents a novel idea of cerebellar computation, which could be of fundamental importance for the cerebellar research community. The authors have clarified a number of important details in the revision and the simulation and analysis methods are solid. Overall, the nature of the paper remains somewhat speculative, as the modelling effort is not linked very tightly to experimental data and other models are not ruled out very convincingly.